# Temporal cascade of frontal, motor and muscle processes underlying human action-stopping

**Sumitash Jana†\*, Ricci Hannah†\*, Vignesh Muralidharan, Adam R Aron**

Department of Psychology, University of California, San Diego, United States

**Abstract** Action-stopping is a canonical executive function thought to involve top-down control over the motor system. Here we aimed to validate this stopping system using high temporal resolution methods in humans. We show that, following the requirement to stop, there was an increase of right frontal beta (~13 to 30 Hz) at ~120 ms, likely a proxy of right inferior frontal gyrus; then, at 140 ms, there was a broad skeletomotor suppression, likely reflecting the impact of the subthalamic nucleus on basal ganglia output; then, at ~160 ms, suppression was detected in the muscle, and, finally, the behavioral time of stopping was ~220 ms. This temporal cascade supports a physiological model of action-stopping, and partitions it into subprocesses that are isolable to different nodes and are more precise than the behavioral latency of stopping. Variation in these subprocesses, including at the single-trial level, could better explain individual differences in impulse control.

## Introduction

The ability to control one's actions and thoughts is important for our daily lives; for example: changing gait when there is an obstacle in the path (*Wagner et al., 2016*), resisting the temptation to eat when on a diet (*Sedgmond et al., 2019*), and overcoming the tendency to say something hurtful (*Xue et al., 2008*). While many processes contribute to such forms of control, one important process is response inhibition – the prefrontal (top-down) stopping of initiated response tendencies (*Aron, 2007*). In the laboratory, response inhibition is often studied with the stop-signal task (*Verbruggen et al., 2019*). On each trial, the participant initiates a motor response, and then, when a subsequent Stop signal occurs, tries to stop. From the behavioral data one can estimate a latent variable; the latency of stopping known as Stop Signal Reaction Time (SSRT), which is typically 200–250 ms in healthy adults (*Verbruggen et al., 2019*). SSRT has been useful in neuropsychiatry where it is often longer for patients vs. controls (*Alderson et al., 2007*; *Bari and Robbins, 2013*; *Lavagnino et al., 2016*; *Lijffijt et al., 2005*; *Smith et al., 2014*; *Snyder et al., 2015*). The task has also provided a rich test-bed, across species, for mapping out a putative neural architecture of prefrontal-basal-ganglia-regions for rapidly suppressing motor output areas (*Aron et al., 2014*; *Bari and Robbins, 2013*; *Schall and Godlove, 2012*). Given this rich literature, this task is one of the few paradigms included in the longitudinal Adolescent Brain Cognitive Development study (*Casey et al., 2018*) of 10,000 adolescents over 10 years.

Against this background, a puzzle is that the relation between SSRT and 'real-world' self-reported impulsivity is often weak (*Chowdhury et al., 2017*; *Enkavi et al., 2019*; *Friedman and Miyake, 2004*; *Lijffijt et al., 2004*; *McLaughlin et al., 2016*; *Skippen et al., 2019b*). One explanation is that SSRT may not accurately index the brain's true stopping latency. Indeed, recent mathematical modelling of behavior during the stop-signal task suggests that standard calculations of SSRT may overestimate the brain's stopping latency by ~100 ms (*Skippen et al., 2019b*; also see *Bissett and Poldrack, 2019*). Further, in a recent study (*Raud and Huster, 2017*), electromyographic (EMG)

**\*For correspondence:**
s2jana@ucsd.edu (SJ);
rhannah@ucsd.edu (RH)

†These authors contributed equally to this work

recordings revealed an initial increase in EMG activity in response to the Go cue, followed by a sudden decline at ~150 ms after the Stop signal. This decline in EMG could be because of the Stop process 'kicking in' to cancel motor output – but the striking thing is that this was 50 ms before the SSRT of 200 ms. This timing is also consistent with experiments using transcranial magnetic stimulation (TMS) to measure the motor evoked potential (MEP) during the stop-signal task (the MEP indexes the excitability of the pathways from motor cortex to muscle). The MEP in the muscle that was-to-be-stopped reduced at ~150 ms (*Coxon et al., 2006*; *van den Wildenberg et al., 2010*). Further, other studies that measured the MEP from muscles that were not needed for the task, show there is 'global suppression' also at ~150 ms (*Badry et al., 2009*; *Cai et al., 2012*; *Wessel et al., 2013a*; *Wessel and Aron, 2013*) (i.e. corticospinal activity was suppressed for the broader skeleto-motor system). This 'global MEP suppression' has been linked to activation of the subthalamic nucleus of the basal-ganglia (*Wessel et al., 2016*), which is thought to be critical for stopping, and might broadly inhibit thalamocortical drive (*Wessel and Aron, 2017*).

The potential overestimation of the brain's true stopping latency by SSRT could arise for several reasons. First, the race model assumes that the Stop process is 'triggered' on every trial. But recent research shows that this is not the case (*Skippen et al., 2019b*), and that failing to account for 'trigger failures' inflates SSRT. Second, while the standard 'race model' assumes that the Go and Stop processes are independent (*Verbruggen et al., 2019*), recent research show that violations of this independence underestimates SSRT (*Bissett and Poldrack, 2019*). Finally, the standard ways of computing SSRT likely do not account for electromechanical delays between muscle activity and the response. In any event, overestimating the brain's stopping latency would add variance to SSRT which could potentially weaken the above-mentioned across-participant associations between stopping latency and self-report scores (*Chowdhury et al., 2017*; *Lijffijt et al., 2004*; *Skippen et al., 2019b*). Furthermore, if the true stopping latency is ~150 ms, the timing of activation of nodes in the putative response inhibition network should precede this time-point for those nodes to play a causal role in action stopping – and this is important for the interpretation of neuroscience studies. For instance, in electrocorticography, electroencephalography (EEG), and magnetoencephalography (MEG) studies, successful stopping elicits increased beta band power over right frontal cortex in the time period between the Stop signal and SSRT (*Castiglione et al., 2019*; *Schaum et al., 2020*; *Swann et al., 2009*; *Wagner et al., 2018*; *Wessel et al., 2013b*). Whether this, and other, neurophysiological markers of the Stop process occur sufficiently early to directly contribute to action-stopping (if SSRT is overestimated) is unknown; yet this is fundamental to our understanding of brain networks underlying response inhibition.

Here we leveraged the insight from the above-mentioned study (*Raud and Huster, 2017*) which used EMG of the task relevant muscles. We now tested whether we could derive a single trial estimate of stopping latency from EMG (referred to as CancelTime). More specifically, we hypothesized that 'partial' EMG bursts on the Successful Stop trials (*i.e.* small EMG responses that begin but do not reach a sufficient amplitude to lead to an overt response) (*de Jong et al., 1990*; *McGarry et al., 2000*) would carry information about the latency of stopping. We tested this in two studies. In a third study we tested if CancelTime would correspond with the measure of putative basal ganglia-mediated global motor suppression, measured with single-pulse TMS. In studies four and five we turned to the cortical process thought to initiate action–stopping, using the above-mentioned proxy of right frontal beta (*Swann et al., 2009*; *Wagner et al., 2018*). We measured scalp EEG, derived a right frontal spatial filter in each participant, and then extracted beta bursts (*Little et al., 2018*) in the time period between the Stop signal and SSRT. We tested how the timing of these beta bursts related to CancelTime.

## Results

### Study 1 (EMG)

10 participants performed the stop-signal task (*Figure 1a*). On each trial they initiated a manual response when a Go cue occurred, and then had to try to stop when a Stop signal suddenly appeared on a minority of trials. Depending on the stop signal delay, SSD, participants succeeded or failed to stop, each ~50% of the time). We measured EMG from the responding right index and little fingers (*Figure 1b* inset). Behavioral performance was typical, with SSRT (referred to as

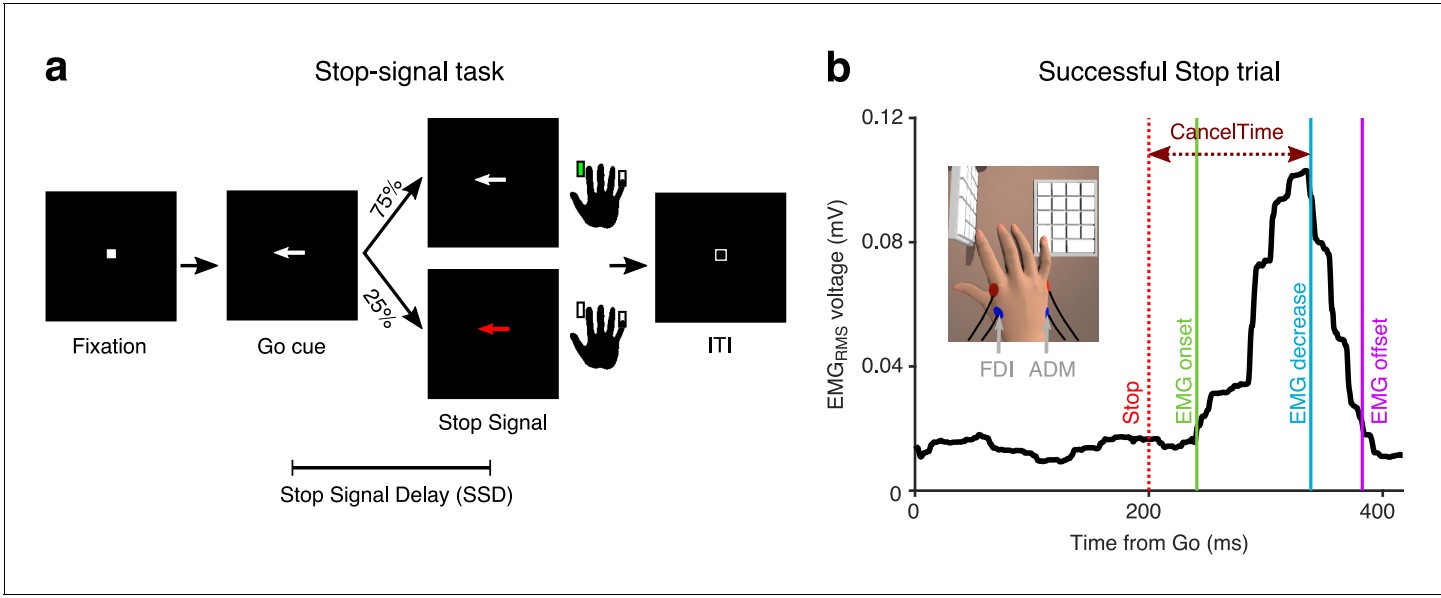

**Figure 1.** Behavioral task and EMG recording. (**a**) Stop-signal task. (**b**) EMG_RMS on a Successful Stop trial (Partial EMG) in an exemplar participant. Data are aligned to the Go cue. CancelTime refers to the time from the Stop signal (dotted red line) to when the EMG_RMS starts decreasing (blue line). The green and purple line represent the detected onset and offset of the EMG response. (*Inset*) Recording set-up with a vertical and a horizontal keypad to record keypresses from the FDI and ADM muscles.

The online version of this article includes the following source data and figure supplement(s) for figure 1:

**Figure supplement 1.** EMG responses in study 1 and 2.

**Figure supplement 1—source data 1.** Mean RT_EMG and RT_Beh in the Correct Go and Failed Stop trials for study 1 and 2.

SSRT_Beh) of 216 ± 8 ms, and action-stopping on 51 ± 1% of Stop trials (*Table 1*). EMG analysis was performed on the trial-by-trial root-mean-squared EMG (EMG_RMS; *Figure 1b*). On 53 ± 6% of Successful Stop trials (*i.e.* where no keypress was made) there was a small but detectable EMG response (Partial EMG trials; see *Figure 1—figure supplement 1* for RT_EMG-RT_Beh correlation), while on the remainder of Successful Stop trials there was no detectable EMG response (No EMG trials). The amplitude of EMG responses (mean peak EMG voltage) in the Partial EMG trials was 48 ± 3% smaller than in trials with a keypress (*Figure 2a*).

We hypothesized that the time when the Partial EMG response starts declining after the Stop signal is a readout of the time when the Stop process is implemented in the muscle (hereafter 'CancelTime'). We observed that, first, CancelTime is much earlier than SSRT_Beh (see *Figure 2c* (*left*) for all CancelTimes in an exemplar participant; mean CancelTime = 146 ± 3 ms, SSRT_Beh = 203 ms); and second, across participants, CancelTime was positively correlated with SSRT_Beh (*Figure 2d*; study 1:

**Table 1.** Behavior (mean ± s.e.m.; All values in ms).

|  | Study 1 (EMG) | Study 2 (EMG) | Study 3 (TMS) | Study 4 (EEG) | Study 5 (EEG) |
|---|---|---|---|---|---|
| Go RT_Beh | 470 (15) | 493 (15) | 430 (17) | 427 (15) | 405 (6) |
| Failed Stop RT_Beh | 416 (11) | 447 (14) | 391 (12) | 384 (12) | 370 (5) |
| Correct Go % | 97 (1) | 98 (0) | 99 (0) | 99 (0) | 99 (0) |
| Correct Stop % | 51 (1) | 52 (1) | 49 (1) | 48 (1) | 50 (0) |
| Mean SSD | 237 (20) | 280 (17) | 194 (18) | 191 (21) | 170 (7) |
| SSRT_Beh | 216 (8) | 204 (4) | 219 (6) | 214 (9) | 219 (6) |

The online version of this article includes the following source data for Table 1:

**Source data 1.** Behavior in the Stop-signal task in all five studies.

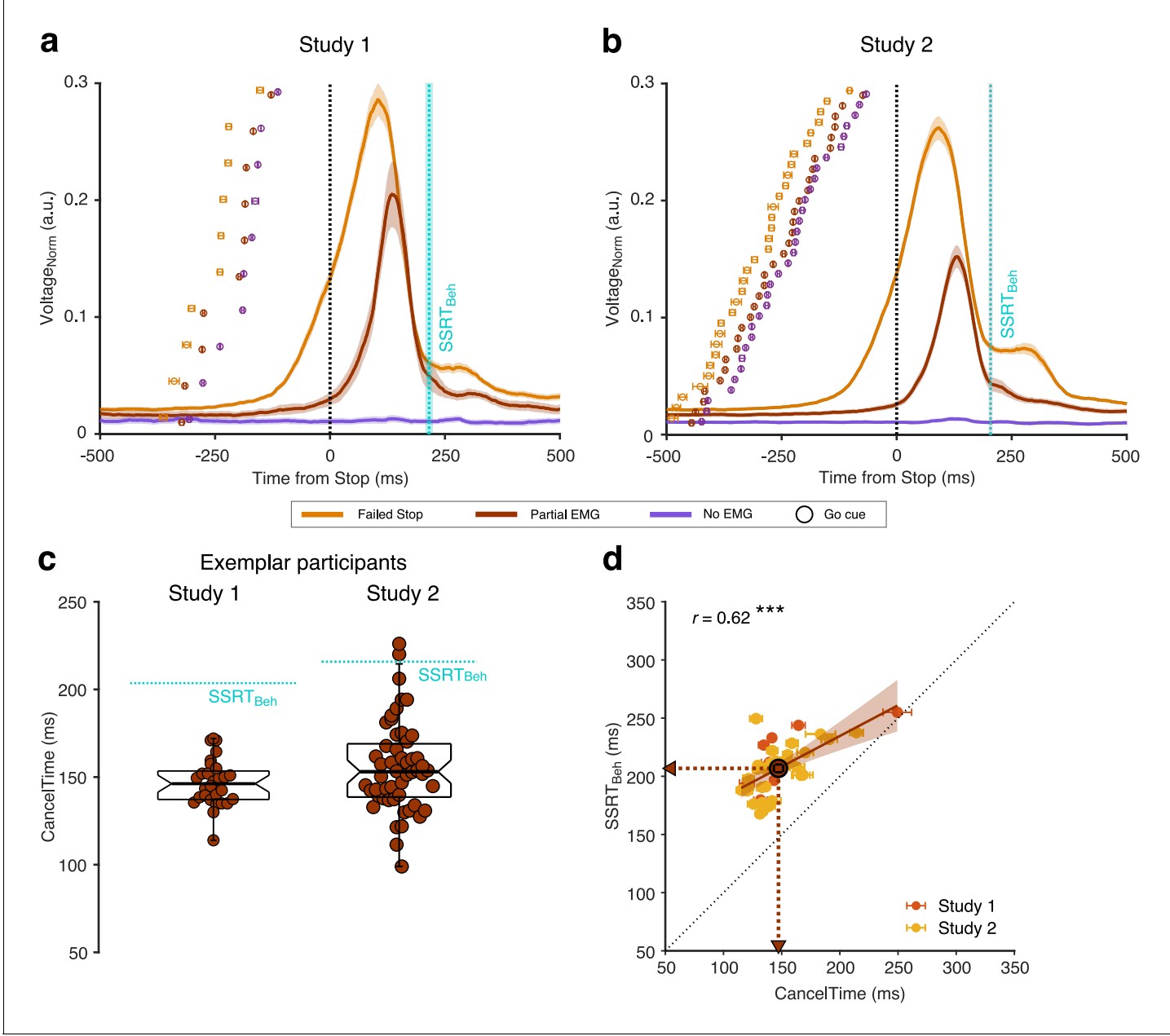

**Figure 2.** EMG responses in Successful (Partial and No EMG) and Failed Stop trials in study 1 and 2. (**a**) Normalized EMG_RMS voltage in Failed Stop (orange), Partial EMG (brown), and No EMG trials (purple), aligned to the Stop signal. The lines and the shaded area represent the mean ± s.e.m. across participants. The dotted cyan line and shaded area represent the mean ± s.e.m of SSRT_Beh across participants. The dots and cross-hairs represent the mean ± s.e.m. of the Go cue in a participant. Note that the time between the Go cue and the Stop signal (i.e. the SSD) is shortest for the No EMG (purple), then the Partial EMG (brown), and then the Failed Stop trials (orange). (**b**) Same as (a) but for study 2. (**c**) (*Left*) Beeswarm plot of the CancelTime in an exemplar participant from study 1. Each dot represents a trial. The dotted cyan line represents the SSRT_Beh. (*Right*) Same as *left* but for study 2. (**d**) Correlation between CancelTime and SSRT_Beh in study 1 (light red) and study 2 (yellow). The brown dot, lines and arrows represent the means, while the black dotted line represents the unity line. The linear regression fit and its 95% confidence interval (pooled study 1 and 2) is shown as a brown line and shaded region respectively.

The online version of this article includes the following source data for figure 2:

**Source data 1.** Correlation between CancelTime and SSRTBeh for study 1 and 2.

mean CancelTime = 152 ± 11 ms, mean $SSRT_{Beh}$ = 216 ± 8 ms; $r$ = 0.71, p=0.020, $BF_{10}$ = 3.6). This suggests that CancelTime might index the time when stopping is implemented at the muscle.

## Study 2 (EMG)

We then ran a new sample ($n$ = 32; see *Table 1* for behavioral results). Again, we observed partial EMG responses on 49 ± 2% of Successful Stop trials; where the EMG amplitude was 54 ± 1% smaller than the amplitude in trials with a keypress (*Figure 2b*). *Figure 2c* (*right*) shows the distribution of CancelTimes in an exemplar participant (mean CancelTime = 156 ± 4 ms, $SSRT_{Beh}$ = 218 ms). Again, across participants, mean CancelTime was positively correlated with $SSRT_{Beh}$ (*Figure 2d*; mean CancelTime = 146 ± 4 ms, mean $SSRT_{Beh}$ = 204 ± 4 ms; $r$ = 0.59, p<0.001, $BF_{10}$ = 71.7). Intriguingly, in each study, CancelTime was ~ 60 ms less than $SSRT_{Beh}$. To further explore this, we pooled the data across the two studies.

## Pooled studies 1 and 2

Mean CancelTime (147±5 ms) was 60±3 ms shorter than $SSRT_{Beh}$ ($t$(41) = 18.4, $p < 0.001$, $d$ = 2.5, $BF_{10} > 100$; $r$ = 0.62, $p < 0.001$, $BF_{10} > 100$). However, one must note that the criterion for estimating the stopping latency is different for the two measures, CancelTime uses EMG responses, while $SSRT_{Beh}$ uses the keypress responses. Hence, inherent differences in latencies between the two responses might lead to these incompatible measures of stopping latency. We hypothesized that the difference between $SSRT_{Beh}$ and the time of EMG cancellation (CancelTime) is due to an inherent "ballistic stage" in movements and once the muscle activity crosses the point-of-no-return they can no longer be stopped and a movement is inevitable (*de Jong et al., 1990*; *Mirabella et al., 2006*; *Osman et al., 1986*; *Verbruggen and Logan, 2009*). [The duration of such ballistic stages has been estimated to be ~15 ms in saccades in non-human primates (*Boucher et al., 2007*; *Kornylo et al., 2003*; *Purcell et al., 2010*) and ~50 ms for reaching movements in humans (*Gopal and Murthy, 2016*; *Jana and Murthy, 2018*)]. In other words, the time of EMG cancellation on partial trials reflects a time just before the point-of-no-return, whereby if EMG activity is allowed to continue develop beyond this point it will exceed a critical threshold such that a button press necessarily ensues (we presume this threshold reflects the point at which the inertia of the finger is overcome). In this respect, what is being tracked by the SSD staircasing procedure is the probability of crossing that EMG threshold, but since SSRT is calculated based on button press response times, it inevitably incorporates the ballistic stage that follows the crossing of this threshold. Hence, although our study was not designed to track the SSD staircase based on EMG, we calculated SSRT using the presence of EMG responses ($SSRT_{EMG}$) instead of the keypress responses ($SSRT_{Beh}$). The purpose of the $SSRT_{EMG}$ estimation was to test the idea of a ballistic phase by removing the influence of electromechanical delays and inertia in the neuromuscular system and response device, which likely make-up the ballistic stage, on the estimated stopping latency. We thus considered Partial EMG trials as Failed Stop trials and used EMG onset time ($RT_{EMG}$) on Correct Go trials to recalculate SSRT (*i.e.* instead of using P(Respond|Stop) from behavior and Go $RT_{Beh}$ as is typical for $SSRT_{Beh}$ calculations; see Materials and methods; see *Figure 3a* for an exemplar participant). We then performed 1-way repeated measures ANOVA with 'Stop Time' as the dependent measure and the method of estimation as a factor ($SSRT_{EMG}$, $SSRT_{Beh}$, and CancelTime). There was a significant main effect of the estimation method on 'Stop Time' ($F_{GG}$(1.4, 56.1) = 66.3, $p < 0.001$, $\eta_p^2$ = 0.6). Pairwise comparisons showed that $SSRT_{EMG}$ (157±7 ms) was significantly faster than $SSRT_{Beh}$ (207±3 ms) (*Figure 3b*; $t$(41) = 8.2, $p_{Bon} < 0.001$, $d$ = 1.3, $BF_{10} > 100$), but importantly, not significantly different from mean CancelTime ($t$(41) = 1.5, $p_{Bon}$ = 0.270, $d$ = 0.2, $BF_{10}$ = 0.5). This suggests that $SSRT_{Beh}$ might be protracted by a peripheral delay and that CancelTime might be a better metric of the time of implementation of the Stop process. [Our simulations using a previously described modelling framework (*Boucher et al., 2007*; *Ramakrishnan et al., 2012*; *Usher and McClelland, 2001*) also lead credence to this idea, demonstrating that the duration of the ballistic stage might be ~35 ms or longer (see *Figure 3—figure supplement 1* and Appendix 1)].

Next, we examined in more detail the EMG profile on Partial EMG trials. Across all participants, the EMG response in the Partial EMG trials (when aligned to the EMG onset) had a profile similar to the EMG response in the Correct Go and Failed Stop trials, but diverged ~55 ms after EMG onset (55 ms compared to Correct Go, and 56 ms compared to Failed Stop trials, *Figure 3c*). We surmised

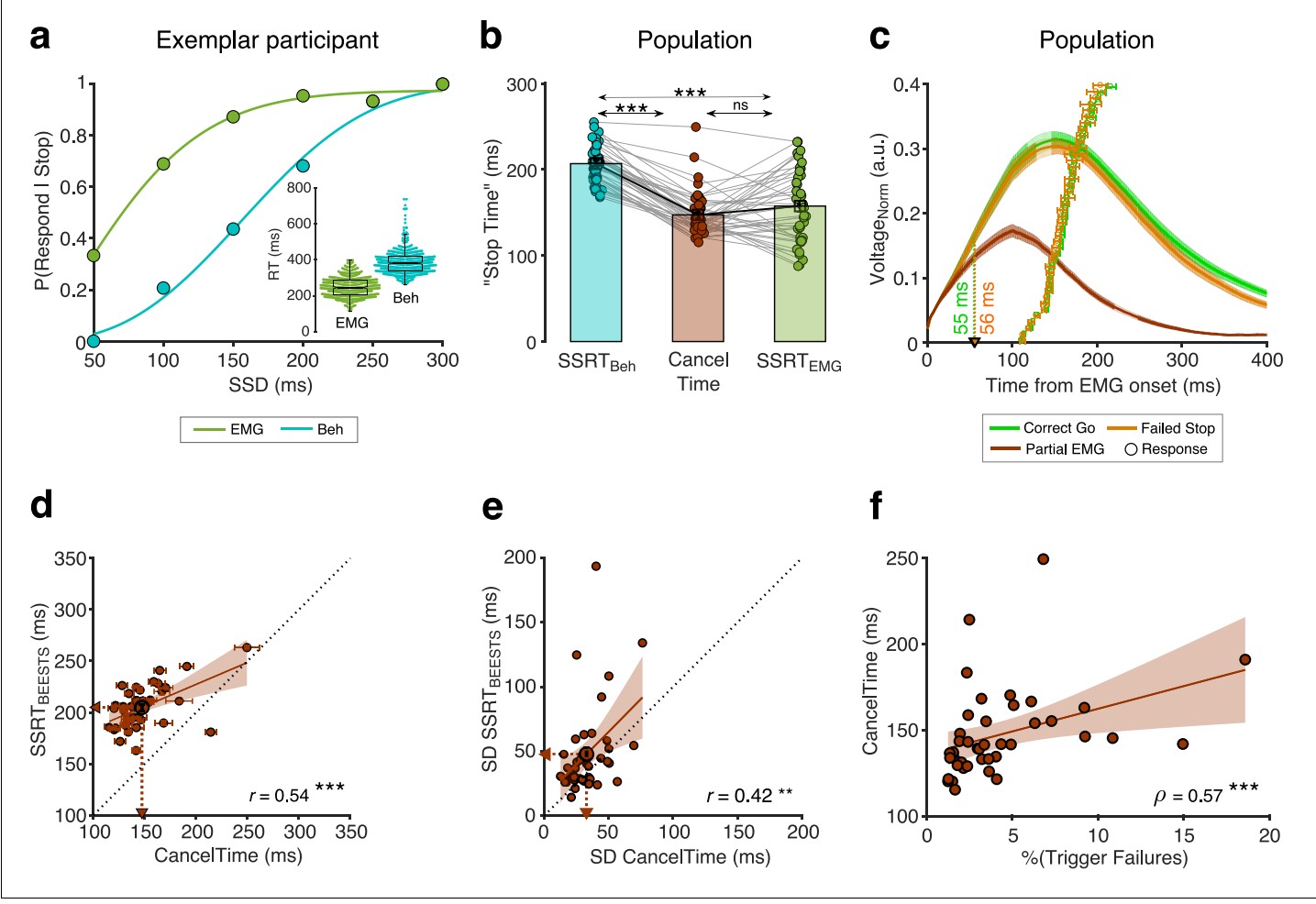

**Figure 3.** Peripheral delay associated with $SSRT_{Beh}$ and the relationship between CancelTime and BEESTS parameters. (a) P(Respond|Stop) in an exemplar participant calculated using the behavioral response (dark green dots) and the EMG response (cyan dots). The lines represent the cumulative Weibull fit as $w(t) = \gamma - (\gamma - \delta)e^{[-(t/\alpha)\beta]}$ where $t$ is the SSD, $\alpha$ is the time at which the function reaches 64% of its full growth, $\beta$ is the slope, $\delta$ is the minimum value of the function, and $\gamma$ is maximum value of the function. The difference between $\delta$ and $\gamma$ marks the range of the function. (*Inset*) Beeswarm plot of the EMG onset (dark green) and the behavioral responses (cyan) used to calculate $SSRT_{EMG}$ and $SSRT_{Beh}$ respectively. (b) Comparison of $SSRT_{Beh}$ (cyan), CancelTime (brown), and $SSRT_{EMG}$ (dark green) across all participants. Each dot represents a participant, while the bar and cross-hair represents the mean ± s.e.m. in a group. (c) The normalized EMG responses aligned to the detected EMG onsets in the Correct Go (green), Failed Stop (orange), and Partial EMG (brown) trials. The line and shaded region represent the mean ± s.e.m. in a group. The dots and cross hairs represent the mean ± s.e.m. of the keypress in a participant. (d) Correlation between CancelTime and mean $SSRT_{BEESTS}$ estimate. Each dot and cross-hair represent the mean ± s.e.m. in a participant. The brown line and the shaded area represent the linear regression fit and its 95% confidence interval. The unity line is represented as a dotted black line. (e) Correlation between SD of CancelTime and SD of the $SSRT_{BEESTS}$ estimate. Other details same as (d). (f) Correlation between percentage Trigger Failures estimated from BEESTS and CancelTime. Other details same as (d).

The online version of this article includes the following source data and figure supplement(s) for figure 3:

**Source data 1.** Peripheral delay associated with $SSRT_{Beh}$ and relationship between CancelTime and BEESTS parameters.
**Figure supplement 1.** Simulation results.
**Figure supplement 1—source data 1.** Simulation results.
**Figure supplement 2.** Partial EMG voltage across SSDs in study 1 and 2.
**Figure supplement 2—source data 1.** Partial EMG voltage across SSDs for study 1 and 2.

that if the Partial EMG trials reflect responses that have been actively cancelled at the muscle-level, then the amplitude of these responses should increase with SSD. The rationale was that, at shorter SSDs, the Go process will have been active for a shorter duration, meaning EMG activity will not have increased much before being inhibited, while at longer SSD, the Go process will have been active for a longer duration, meaning EMG activity will have increased much more before being

inhibited. Indeed, the amplitude of the Partial EMG responses increased with SSD (*Figure 3—figure supplement 2*). A 1-way repeated measures ANOVA with amplitude as the dependent variable and the SSD as the independent variable showed significant effect of SSD on amplitude ($F(4,24)$ = 3.7, $p$ = 0.018, $\eta_p^2$ = 0.4) (also see *Coxon et al., 2006*). This suggests that the Partial EMG trials represent inhibited Go responses and not merely a weak Go process (which would presumably not increase across SSDs). In other words, the partial EMG response does not simply reflect a weak Go response, where the individual intended to execute a response but failed to produce sufficient muscle activity to fully depress the button, since the amplitude of such responses would presumably not vary as a function of SSD.

To further validate CancelTime, we modelled the behavior using BEESTS (Bayesian Estimation of Ex-gaussian STop-Signal reaction time distributions; see *Table 2* for model estimates). While SSRT$_{Beh}$ produces a single estimate per person, BEESTS uses a Bayesian parametric approach to estimate the distribution of SSRTs (*Matzke et al., 2017*). Also, for each participant, it provides an estimate of the probability of trigger failures (*i.e.* stop trials where the stopping process was not initiated *Matzke et al., 2017*). Across participants, mean CancelTime was positively correlated with the mean SSRT$_{BEESTS}$ (205 ± 3 ms; $r$ = 0.54, p<0.001, $BF_{10}$ > 100; *Figure 3d*). More interestingly, the SD of CancelTime (33 ± 2 ms) was positively correlated with the SD of SSRT$_{BEESTS}$ (48 ± 5 ms; $r$ = 0.42, p=0.005, $BF_{10}$ = 6.9; *Figure 3e*). Further, the percentage of trigger failures (4 ± 1%) was positive correlated with mean CancelTime ($\rho$ = 0.57, p<0.001, $BF_{10}$ > 100) suggesting that participants who fail to 'trigger' the Stop process more often, are also likely to have longer stopping latency, indicating that there might exist a dependency between the triggering and the implementation of the Stop process (*Figure 3f*). These relationships between CancelTime and model estimates give further credence to our interpretation that CancelTime on Partial EMG trials reflects a single-trial measure of the time of implementation of the Stop process.

## Study 3 (TMS)

To further validate CancelTime and relate it to brain processes we turned to a different method – single-pulse TMS over a task-irrelevant muscle representation in the brain. As mentioned above, the reduction of MEPs from task-irrelevant muscles on Successful Stop trials (*Badry et al., 2009*; *Cai et al., 2012*; *Wessel and Aron, 2013*), is thought to reflect a basal ganglia-mediated global suppression (*Wessel et al., 2016*). Seventeen new participants (see *Table 1* for behavioral results) now performed the task with their left hand, while TMS was delivered over the left motor cortex and MEPs were recorded from a task-irrelevant, right forearm muscle. MEPs were recorded at different times after the Stop signal on different trials: 100–180 ms in 20 ms intervals, as well as during the inter-trial interval which served as a baseline. Concurrently, we recorded EMG from the task-relevant left-hand muscles as in studies 1 and 2 above (*Figure 4a*).

The key TMS finding, in keeping with earlier studies (*Badry et al., 2009*; *Cai et al., 2012*; *Wessel and Aron, 2013*), was of suppression of MEPs in the task-irrelevant forearm, indicating global motor system suppression, beginning ~ 140 ms following the Stop signal in Successful Stop trials (*Figure 4b*; see *Figure 4—figure supplement 1* for MEP amplitudes for Partial EMG and No EMG trials separately). A 2-way repeated measures ANOVA with MEP amplitude as the dependent measure and the factors of trial-type (Correct Go, Successful Stop, Failed Stop) and time (100, 120,

**Table 2.** BEESTS estimates (mean ± s.e.m.; All values in ms)

| Estimated parameters | Pooled study 1 and 2 |
| --- | --- |
| Mean Go RT$_{Beh}$ | 483 (13) |
| SD Go RT$_{Beh}$ | 94 (5) |
| Mean SSRT$_{BEESTS}$ | 205 (3) |
| SD SSRT$_{BEESTS}$ | 48 (5) |
| %Trigger Failures | 4 (1) |

The online version of this article includes the following source data for Table 2:
**Source data 1.** BEESTS estimates for study 1 and 2.

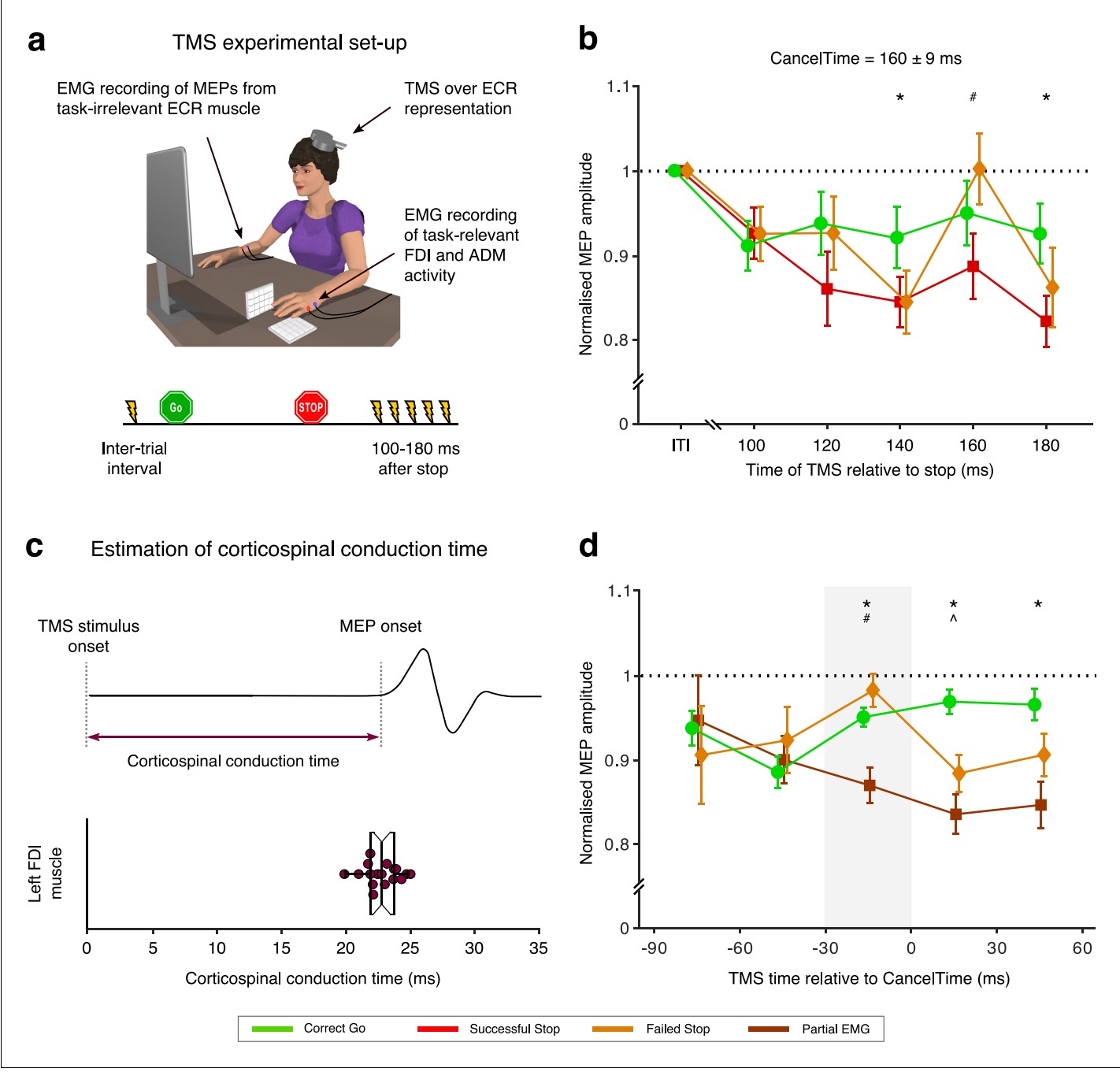

**Figure 4.** Relationship between global motor system suppression and CancelTime. (**a**) Experimental set up and TMS stimulus timings for study 3. Participants performed the Stop signal task with the left hand with concurrent EMG measurement of CancelTime from task-relevant FDI and ADM muscles. On a given trial, a single TMS stimulus over left M1 was delivered at one of 6 possible times to elicit a motor evoked potential (MEP) in the task-irrelevant extensor carpi radialis (ECR) muscle of the right forearm. (**b**) Global motor system suppression begins at 140 ms after the Stop-signal, and thus ~20 ms prior to the mean CancelTime. Paired t-tests: *, $p_{Bon} < 0.05$ Successful Stop (red; combined Partial and No EMG trials) vs. Correct Go (green); #, $p_{Bon} < 0.05$ Successful Stop vs. Failed Stop (orange). Each dot and cross-hairs represent the mean ± s.e.m. across the population. The black dotted line shows amplitude of MEPs normalized to those at the inter-trial interval. (**c**) (*Top*) Schematic representation of an MEP. (*Bottom*) Beeswarm plot of the mean corticospinal conduction time to a hand muscle, which was established by measuring the onset latency of MEPs in the hand (~23 ms on average). Each dot represents a participant. This conduction time is included in CancelTime. (**d**) Trial-by-trial analysis of MEP amplitudes organized into 30 ms time bins reflecting the time of TMS expressed relative to CancelTime. Global motor system suppression begins in a window 30-0 ms prior to the CancelTime (gray shaded region). Wilcoxon rank sum test: *, $p_{Bon} < 0.05$ Partial EMG (brown) vs. Correct Go (green); #, $p_{Bon} < 0.05$ Partial EMG vs. Failed Stop (orange); ˆ, $p_{Bon} < 0.05$ Failed Stop vs. Correct Go. The black dotted line shows amplitude of MEPs normalized to those at the inter-trial interval.

*Figure 4 continued on next page*

*Figure 4 continued*

The online version of this article includes the following source data and figure supplement(s) for figure 4:

**Source data 1.** Relationship between global motor system suppression and CancelTime.
**Figure supplement 1.** MEPs in study 3.

140, 160, 180 ms after the Stop signal) showed main effects of both trial-type ($F(2,32)$ = 7.2, p=0.003, $\eta_p^2$ = 0.3) and time ($F_{GG}(2.5, 40.7)$=4.8, p=0.008, $\eta_p^2$ = 0.2), as well as an interaction of trial-type by time ($F(8, 128)$=3.4, p=0.002, $\eta_p^2$ = 0.2). Post hoc *t*-tests across Successful Stop and Correct Go trials showed *no* difference at 100 ms ($t(16)$ = 0.7, $p_{Bon}$ = 1.0, $BF_{10}$ = 0.3), 120 ms ($t(16)$ = 2.5, $p_{Bon}$ = 0.066, $BF_{10}$ = 2.8), and 160 ms ($t(16)$ = 2.1, $p_{Bon}$ = 0.159, $BF_{10}$ = 1.4). However, MEP amplitudes *were* significantly suppressed on Successful Stop trials at 140 ms ($t(16)$ = 4.1, $p_{Bon}$ = 0.003, $BF_{10}$ = 39.8) and 180 ms ($t(16)$ = 4.4, $p_{Bon}$ < 0.001, $BF_{10}$ = 65.2) after the Stop signal. Therefore, we estimate the onset of the global motor suppression to be ~ 140 ms after the Stop signal, which places it ~ 20 ms prior to the mean CancelTime (160 ± 9 ms). There were no significant differences in MEP amplitudes between Failed Stop and Correct Go trials at any time point, though MEP amplitudes on Successful Stop trials were also suppressed compared to Failed Stop trials at 160 ms ($t(16)$ = 2.9, $p_{Bon}$ = 0.033, $BF_{10}$ = 4.9).

It makes sense that global motor suppression occurs before CancelTime as motor cortical output takes time to be transmitted along the corticospinal pathway to the muscles. To verify whether the ~20 ms discrepancy in timings could be accounted for by corticospinal conduction delays, we estimated this corticospinal conduction time in a separate phase of the current study by delivering TMS over the hand representation to evoke MEPs in the left, task-relevant, FDI muscle (*Figure 4c*). This was 23 ± 0.3 ms. Thus, a decline in muscle activity would be expected to be preceded by a reduction in motor cortical output by ~23 ms, which is very similar to the ~20 ms difference between global motor suppression and CancelTime. Note, however, that the onset latency of the TMS-evoked MEP is likely an under-estimate of the mean conduction time of all pathways involved in voluntary movement, because TMS is biased towards recruiting fast conducting corticospinal neurons with mono-synaptic connections to the spinal motorneurons (*Day et al., 1989*; *Edgley et al., 1997*). Therefore, the mean latency at which changes in motor cortical output are observable as changes in EMG activity is probably longer than 23 ms by several milliseconds.

To further elaborate the temporal relationship between global motor suppression and CancelTime, we performed a trial-by-trial analysis whereby MEP amplitudes were sorted according to the time at which TMS was delivered, relative to the time at which EMG decreased on Successful Stop, Failed Stop and Correct Go trials (*Figure 4d*). The suppression of MEPs in Successful Stop trials compared to Correct Go trials began in the 30 ms prior to the EMG decline (−30 to 0 ms: $Z$ = 3.12, $p_{Bon}$ = 0.005; 0 to 30 ms: $Z$ = 4.48, $p_{Bon}$ <0.001; 30 to 60 ms: $Z$ = 2.45, $p_{Bon}$ = 0.045). This lag in the time of EMG decrease relative to the time of the MEP suppression on Successful Stop trials can again be accounted for by the corticospinal conduction time. Thus, these results imply that the brain output to task-relevant muscles declines at approximately the same time as the global motor suppression begins. We note too, that MEPs were also suppressed in Failed Stop versus Correct Go trials, but at some delay relative to Successful Stops and the time of EMG cancellation (*Figure 4d*). This is consistent with the idea that the Stop process is initiated even in Failed Stop trials, and that part of the reason for the failure to stop is that the Stop process is initiated/implemented later in these trials (the other reason being that the Go process might have been completed particularly quickly).

## Study 4 (EEG)

Having established that CancelTime reflects the time of an active stopping process at the muscle (studies 1 and 2, EMG/behavior), which also related tightly with the timing of global motor suppression (study 3, TMS), we then tested whether this EMG measure was also related to the timing of a prefrontal correlate of action-stopping, specifically the increase of beta power (13–30 Hz) before $SSRT_{Beh}$ at right frontal electrode sites (*Castiglione et al., 2019*; *Wagner et al., 2018*). We now measured scalp EEG as well as EMG from the hand, in 11 participants (see *Table 1* for behavioral

results). We derived beta bursts rather than beta power per se, as bursts have richer features (*Shin et al., 2017*) also see *Little et al. (2018)*; *Wessel (2020)*, such as burst timing and duration.

To identify right frontal electrodes of interest in each participant (i.e. a spatial filter), we used Independent Components Analysis (*Bell and Sejnowski, 1995*; see *Castiglione et al., 2019*; *Wagner et al., 2018*). We selected a participant-specific independent component (IC) based on two criteria; First, the scalp topography (right-frontal, and if not present, frontal); and Second, an increase in beta power on Successful Stop trials (from Stop signal to $SSRT_{Beh}$; $Stop_{Win}$) compared to activity *prior to the Go* cue [−1000 to −500 ms aligned to the Stop signal; see Materials and methods; *Figure 5—figure supplement 1*]. The average scalp topography across all participants is shown in *Figure 5b*, *inset* (see *Figure 5—figure supplement 2* for average dipoles). For each participant, we estimated beta bursts; First, by filtering the data at the peak beta frequency; and Second, by defining a burst threshold based on the beta amplitude in a baseline period *after* the Stop signal (500–1000 ms after Stop signal in the Stop trials, and 500–1000 ms after the mean SSD in the Correct Go trials) (see Materials and methods; *Figure 5—figure supplement 3*).

In an exemplar participant, the burst % increased for Successful Stop compared to both Failed Stop and Correct Go trials prior to $SSRT_{Beh}$ (*Figure 5a*). To quantify this across participants, we compared the mean burst % among the three trial-types, and for the time window from the Stop signal to the $SSRT_{Beh}$ of a participant ($Stop_{Win}$) and the baseline period *before the Stop* signal ($Base_{Win}$; Go to Stop signal in Stop trials and Go to mean SSD in Correct Go trials). We performed a 2-way repeated measures ANOVA with mean burst % as the dependent measure, with trial-type (Successful, Failed Stop, and Correct Go trials) and time-window ($Stop_{Win}$ and $Base_{Win}$) as factors. There was a significant main effect of trial-type ($F(2,20) = 4.5$, p=0.025, $\eta_p^2 = 0.3$) and a trial-type by time-window interaction ($F(2,20) = 4.0$, p=0.034, $\eta_p^2 = 0.3$), but no main effect of time-window ($F(1,10) = 3.8$, p=0.088, $\eta_p^2 = 0.3$). Post hoc *t*-tests showed that in the $Stop_{Win}$ there was a significant increase in burst % for Successful Stop (14.6 ± 1.7%) compared to both its baseline (9.9 ± 1.7%; $t(10) = 3.3$, $p_{Bon} = 0.022$, $BF_{10} = 7.6$), and Correct Go (9.6 ± 1.3%; $t(10) = 3.7$, $p_{Bon} = 0.015$, $BF_{10} = 11.8$), but not to Failed Stop (10.3 ± 1.6%; $t(10) = 2.1$, $p_{Bon} = 0.198$, $BF_{10} = 1.2$) (*Figure 5b*). Thus, burst % increased for the Successful Stop trials which could not be attributed to post-movement beta rebound (see *Figure 5—figure supplement 4*).

To further clarify the temporal relationship between beta activity and the current EMG measure of action-stopping, we quantified the mean burst time (BurstTime in the $Stop_{Win}$) for each participant. Across participants, the mean BurstTime (115 ± 6 ms) was significantly shorter than mean CancelTime (169 ± 10 ms; $t(10) = 8.2$, p<0.001, $BF_{10} > 100$) and there was also a strong positive relationship between them ($\rho = 0.76$, p=0.006, $BF_{10} = 10.6$; *Figure 5d*; see *Figure 5—figure supplement 5* for correlation between CancelTime and other burst parameters). Further, we show that the observed correlation was not merely an artifact of varying $Stop_{Win}$ across participants (permutation test, p<0.05; see Materials and methods). Thus, these results show that participants with an early frontal beta burst also had an early CancelTime.

## Study 5 (EEG replication)

We ran a new sample of 13 participants (see *Table 1* for behavioral results). As above a right frontal IC was extracted for each participant (average topography *Figure 5c inset*) and the burst % was compared for the three trial-types (Successful Stop, Failed Stop, and Correct Go) in the two time-windows ($Stop_{Win}$ and $Base_{Win}$). Again, a 2-way repeated measures ANOVA with burst % as the dependent measure revealed that there was a significant main effect of trial-type ($F(2,24) = 6.9$, p=0.004, $\eta_p^2 = 0.4$) and a trial-type by time-window interaction ($F(1,12) = 5.8$, p=0.009, $\eta_p^2 = 0.3$; *Figure 5c*). Here there was also a significant effect of time-window on burst % ($F(1,12) = 16.1$, p=0.002, $\eta_p^2 = 0.6$). Post-hoc *t*-tests confirmed that the burst % was greater for Successful Stop (16.2 ± 2.2%) compared to its baseline (11.3 ± 1.4%; $t(12) = 3.3$, $p_{Bon} = 0.021$, $BF_{10} = 7.6$), and Correct Go (12.0 ± 1.4%; $t(12) = 3.0$, $p_{Bon} = 0.030$, $BF_{10} = 5.3$) but not compared to Failed Stop (15.4 ± 1.4%; $t(12) = 1.0$, $p_{Bon} = 0.957$, $BF_{10} = 0.34$). Across participants, the mean BurstTime (129 ± 7 ms) was again significantly shorter than CancelTime (166 ± 8 ms; $t(10) = 5.0$, p<0.001, $BF_{10} > 100$) and there was a significant positive relationship ($\rho = 0.57$, p=0.045, $BF_{10} = 1.9$; *Figure 5d*). Again, a permutation test suggested that this correlation was unlikely to result from mere variation in the length of $Stop_{Win}$ across participants (p<0.05). Combining data from studies 4 and 5 confirms the strong relationship between right frontal beta BurstTime and CancelTime ($\rho = 0.66$, p<0.001, $BF_{10} = 29.4$).

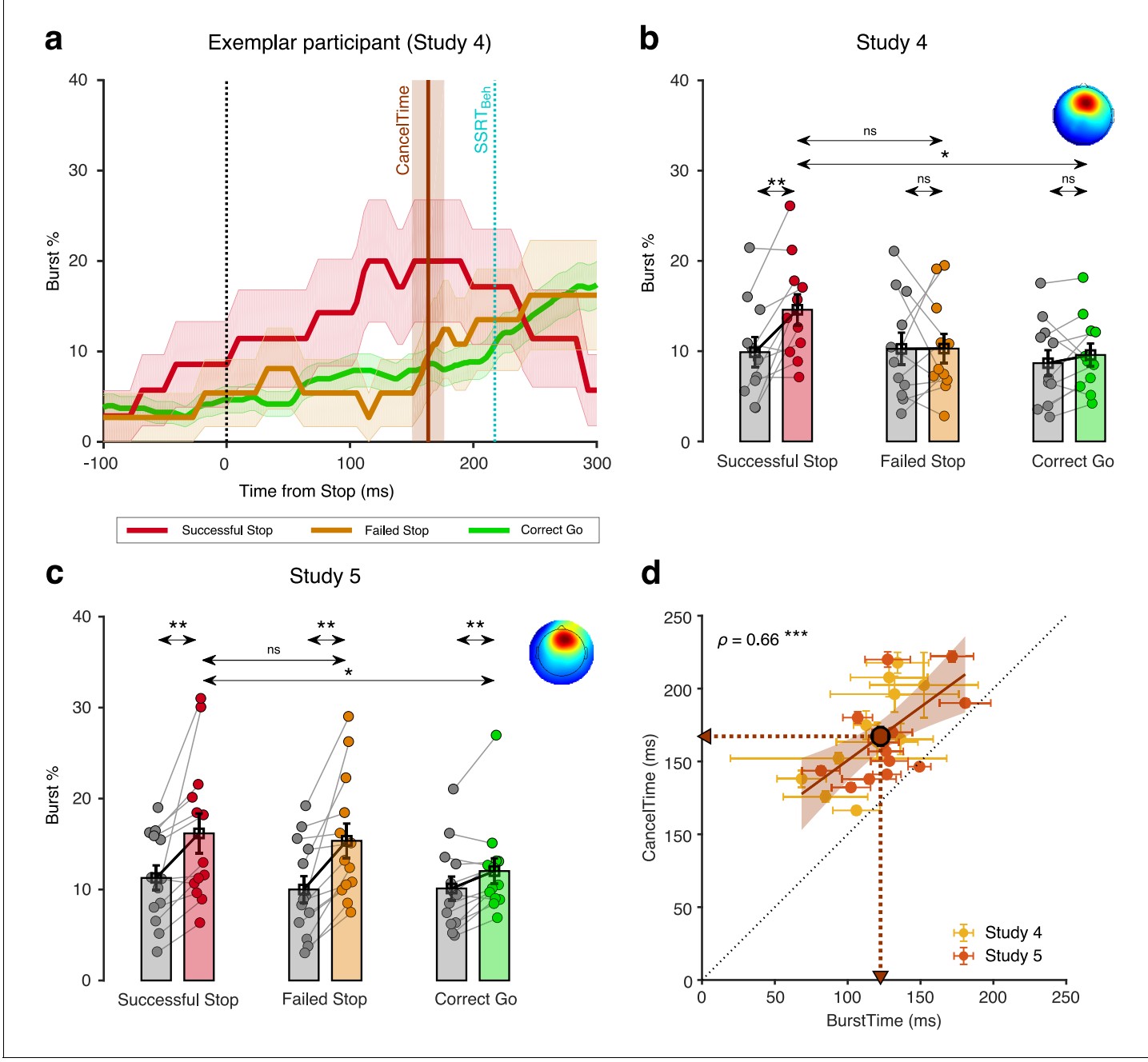

**Figure 5.** Relationship between scalp EEG beta bursts and CancelTime (study 4 and 5). (**a**) Burst % across time for Successful Stop (red), Failed Stop (orange), and Correct Go (green) trials for an exemplar participant in study 4 from the right frontal spatial filter. The shaded region represents mean ± s. e.m. The CancelTime is shown in brown and the SSRTBeh as a cyan line. (**b**) The mean burst probability across all participants for Successful Stop (red), Failed Stop (orange), and Correct Go (green) trials and their respective baselines (gray). The bars and cross-hairs represent the mean and s.e.m across participants, while the dots represent individual participants. (Inset top right) The average scalp topography of all the right frontal ICs across all participants. (**c**) Same as (**b**) but for study 5. (**d**) Correlation between mean BurstTime and mean CancelTime. The yellow dots and cross-hairs represent the participants in study 4, while the light red ones represent participants in study 5. The brown line and the shaded area represent the linear regression fit and its 95% confidence interval (pooled study 4 and 5). Other details same as Figure 2d.

The online version of this article includes the following source data and figure supplement(s) for figure 5:

**Source data 1.** Relationship between scalp EEG beta bursts and CancelTime for study 4 and 5.
**Figure supplement 1.** Beta power in Successful Stop trials for study 4.
**Figure supplement 2.** Average dipole location of the ICs selected in study 4 and 5.
**Figure supplement 3.** Illustration of beta burst computation.

*Figure 5 continued on next page*

*Figure 5 continued*

**Figure supplement 4.** Dynamics of burst% aligned to the Go cue.
**Figure supplement 5.** Relationship of CancelTime with other burst parameters.
**Figure supplement 5—source data 1.** Relationship between CancelTime and other burst parameters in study 4 and 5.

## Discussion

This set of studies provides detailed information about the timing of subprocesses in human action-stopping. We started with the recently published observations that the standard behavioral measure of action-stopping (SSRT) is, an over-estimate of stopping latency (*Bissett and Poldrack, 2019*; *Raud and Huster, 2017*; *Skippen et al., 2019b*). To more precisely delve into this, we validated a trial-by-trial method for estimating stopping latency from EMG. We focused on Successful Stop trials with small impulses (partial bursts) in EMG activity. The amplitude of such partial EMG activity was ~ 50% of the amplitude of EMG activity for outright keypresses, and this decreased at ~160 ms after the Stop signal (CancelTime), which is similar to other studies (*Raud et al., 2019*; *Raud and Huster, 2017*). While, one interpretation of this partial EMG activity is that it merely reflects 'weak' Go activation that did not run to completion (*de Jong et al., 1990*), several lines of evidence strongly suggest it is a muscle manifestation of the stopped response. First, CancelTime was positively correlated with SSRT$_{Beh}$, similar to recent studies (*Huster et al., 2019*; *Thunberg et al., 2019*). Second, the variability of CancelTime was positively correlated with the variability of SSRT estimated from the BEESTS modeling framework. Third, the partial EMG activity had a profile which was initially similar to the EMG profile seen when actual keypresses were made, and only diverged at ~55 ms after EMG onset. This initial similarity would not be expected if it were a weak Go activation – since previous research has demonstrated that weak and strong muscle activations have distinct profiles that diverge soon after onset (*Bellumori et al., 2011*). Fourth, our TMS experiment demonstrated that CancelTime coincided well with the timing of a putative basal ganglia-mediated global motor suppression (*Badry et al., 2009*; *Cai et al., 2012*; *Wessel and Aron, 2013*; *Wessel et al., 2016*; *Wessel and Aron, 2013*; *Wessel and Aron, 2017*). This implies that the smaller amplitude and earlier decline of the partial EMG activity on Successful Stop was due to an active suppression of motor output. Fifth, across participants, on Successful Stop trials, CancelTime correlated strongly with the time of right frontal beta bursts (BurstTime) from scalp EEG. This is consistent with response inhibition being implemented via right prefrontal cortex (*Aron et al., 2014*), and with previous research showing an increase of beta at right frontal electrode sites before SSRT$_{Beh}$ (*Castiglione et al., 2019*; *Wagner et al., 2018*).

Due to the poor spatial resolution of EEG it is not possible to pin down the origin of the bursts recorded on the scalp to any particular frontal cortical area (see *Figure 5—figure supplement 2*) – these bursts could relate to the rIFC or the presupplementary motor area, preSMA, or both [the rIFC and preSMA are connected via the aslant tract (*Catani et al., 2013*; *Swann et al., 2012*)]. We note again that two studies with intracranial EEG showed increases of right frontal beta for rIFC (*Swann et al., 2009*; *Swann et al., 2012*) and also that a recent study using source reconstruction of MEG signals based on fMRI in the same subjects showed an especially strong beta power increase for rIFC, that began ~ 140 ms after the Stop-signal (*Schaum et al., 2020*), consistent with our results.

We also acknowledge that the burst % was quite low on Successful Stop trials (~15%) and that CancelTimes on trials with and without bursts were not different (Study 5: CancelTime$_{With\ Burst}$ = 164 ± 9 ms; CancelTime$_{No\ Burst}$ = 165 ± 9 ms, $t(12)$ = 0.6, p=0.58, $d$ = 0.2, $BF_{10}$ = 0.3; Study 4: too few trials for meaningful comparisons). While this might indicate that bursts are not necessary or sufficient for action-stopping, we think that the poor signal-to-noise of EEG could explain the low burst %. Further research is needed to test if beta bursts are causal to stopping. On a related point, the presence of beta on Go trials was also interesting. It is possible that beta bursts on Go trials reflected the (partly) spontaneous events that occur periodically (but have some functional consequence) (*Shin et al., 2017*), or the bursts might have had a role in proactive slowing on Go trials (as the task, after all, required participants to prepare to stop their response).

While several scalp EEG, intracranial EEG, and MEG studies showed increased right frontal beta power for stopping (*Castiglione et al., 2019*; *Schaum et al., 2020*; *Swann et al., 2009*;

*Swann et al., 2012*; *Wagner et al., 2018*), a recent scalp EEG study focused on the spatial and temporal dynamics of beta bursts (*Wessel, 2020*). That study saw that burst probability increased for likely dorsomedial frontal cortex (electrode FCz) rather than right frontal cortex, as we do. This discrepancy could be explained by our use of a spatial filter approach whereas that study analyzed the data in channel space. A further observation of *Wessel (2020)* was that bursts increased over bilateral sensorimotor cortex ~ 25 ms after the frontal area; and this was interpreted as inhibition of the motor system. This fits our observation of a decrease in corticospinal excitability within ~ 20 ms of right frontal bursts. Putting aside methodological differences, these studies together implicate beta bursts in action-stopping.

A puzzle in our results was that CancelTime was ~ 60 ms earlier than $SSRT_{Beh}$. To better understand this discrepancy, we calculated SSRT based on the EMG response rather than behavior. We saw that $SSRT_{EMG}$ better matched CancelTime than did $SSRT_{Beh}$. Thus, $SSRT_{Beh}$ could be an overestimation of the duration of the Stop process in the brain. This extra time in $SSRT_{Beh}$ probably reflects a 'ballistic stage' in generation of the button press (*de Jong et al., 1990*). We suggest that the maximum CancelTime reflects the last point at which a Stop process can intervene to prevent responses. We note that CancelTime (a muscle measurement) is an overestimation of the brain's stopping latency since it does not include the corticospinal conduction time, which we estimated as ~ 20 ms, and does not include the stopping latencies of the No EMG trials, which presumably reflect the fastest stopping latencies where the Stop process was fast enough to cancel the impending response before it reaches the muscle. Indeed, our TMS results show that global motor suppression, which we take as the time at which motor areas of the brain are suppressed, is ~ 140 ms (which is ~ 20 ms less than CancelTime). One important consequence of our observation that the brain's stopping latency is ~ 140 ms is that neural events that mediate stopping need to occur before this time. Indeed, we found that right frontal beta activity increased ~ 120 ms after the Stop signal on Successful Stop trials, and also that, across participants, there was a strong positive relationship between mean BurstTime and mean CancelTime.

Taken together, these studies motivate a relatively detailed model of the temporal events of action-stopping (*Figure 6*; *Video 1*). First, we suppose the right frontal beta bursts relate to activity of right inferior frontal gyrus (*Aron et al., 2014*; *Swann et al., 2009*), and this happens in ~ 120 ms, which then leads via basal ganglia (*Wessel et al., 2016*) to global suppression of the primary motor cortex (*Badry et al., 2009*; *Cai et al., 2012*; *Wessel and Aron, 2013*; *Wessel and Aron, 2013*; *Wessel and Aron, 2017*) at ~140 ms. After a corticospinal conduction delay of ~ 20 ms, this suppression of motor output is then reflected at ~160 ms as a decline in muscle activity (CancelTime). Finally, $SSRT_{Beh}$ occurs at ~220 ms, after, what we suppose is an electromechanical delay of ~ 60 ms. Thus, CancelTime *narrows* the time window for the causal manipulation of neural structures involved in action-stopping. This is in contrast to previous studies that have proposed that the onset of intramuscularly-recorded antagonist EMG responses (which is *longer* than SSRT) can be used as an alternative for estimating the stopping latency (*Atsma et al., 2018*; *Corneil et al., 2013*; *Goonetilleke et al., 2012*; *Goonetilleke et al., 2010*).

We acknowledge that the timings in this model are approximations that are dependent on a range of factors (such as averaging across participants, running different experiments, and the particular parameters of detection algorithms). However, we note a striking convergence of timings across the current experiments and in other studies, for example CancelTime (*Hannah et al., 2019*; *Raud et al., 2019*; *Raud and Huster, 2017*), time of MEP suppression (*Coxon et al., 2006*; *van den Wildenberg et al., 2010*), corticospinal conduction time (*Groppa et al., 2012*; *Hamada et al., 2013*), BurstTimes (*Hannah et al., 2019*), and the ballistic stage (*Gopal and Murthy, 2016*; *Jana and Murthy, 2018*). Together, these all provide support for our model.

This model specifies the possible chronometrics of stopping in more detail than extant human models, and, more generally, raises questions about the timing reported in some other studies where the neural change appears late. For example, movement neurons in monkey Frontal Eye Field decrease activity in less than 10 ms before SSRT (*Hanes et al., 1998*), dopaminergic neurons in rodent substantia nigra and striatum increase activity only 12 ms before SSRT (*Ogasawara et al., 2018*), TMS at ~ 25 ms before SSRT over human Intraparietal Sulcus prolongs SSRT (*Osada et al., 2019*), and P300 human EEG activity ~ 300 ms after the Stop signal relates to the stopping latency (*Wessel and Aron, 2015*). Whereas the rather late timing of some of these results might be related to processes such as monitoring and feedback (*Huster et al., 2019*) as has been ascribed to brain

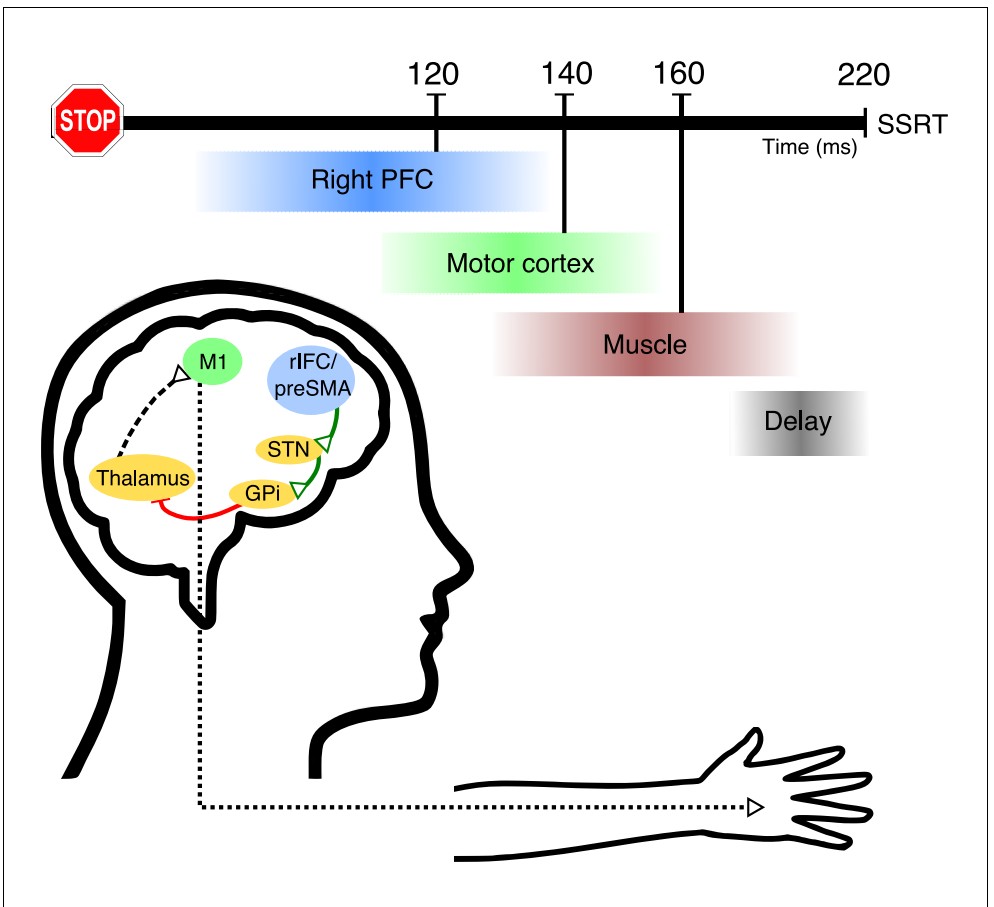

**Figure 6.** Hypothetical model of the temporal cascade of processes underlying human action-stopping. Following the Stop signal, the right PFC including the rIFC and the preSMA gets activated at ~120 ms. These region/s activate (green connections) the STN of the basal ganglia. This in turn activates the globus pallidus interna which, via its inhibition (red connection) on the motor regions of the thalamus, cuts down the 'drive' to the motor cortex. Theoutcome is a global motor suppression at ~140 ms after the Stop signal. This suppression is reflected in the hand muscle at ~160 ms which is measured as the CancelTime. There is a delay of ~60 ms at the muscle level which gets added to the behavioral estimate of SSRT.

signatures that modulate after SSRT (*Logan et al., 2015*; *Schall and Boucher, 2007*), our earlier latencies for prefrontal bursts, TMS-MEP and muscle CancelTime are more indicative of a role in stopping itself.

While our study specifically looked at the chronometrics of the Stop process, and tried to better characterize the physiological model underlying action-stopping, we also now speculate how our results relate to computational models of action-stopping (specifically, the independent race model, the interactive race model, the BEESTS model, and the blocked-input model, and the). First, our results are not compatible with a strictly independent model (of the Go and the Stop processes) since we see active inhibition of M1 (the Go process) already some time before SSRT. Second, our results are compatible with the interactive race model which suggests that the Stop process begins late, but implementation is quick (i.e. within the last 1/3th of the stopping latency) (*Boucher et al., 2007*). Indeed, we observed beta bursts in frontal areas ~ 120 ms after the stop signal followed by rapid cancellation at the muscle within ~ 40 ms. Third, our results are also compatible with the BEESTS model insofar as they point to a trigger process that has a duration of about 80–120 ms (*Bekker et al., 2005*; *Skippen et al., 2019a*). Finally, the interactive-race model and blocked-input model are very similar (*Logan et al., 2015*), so our results do not disambiguate them.

Our results have several important implications. First, whereas several earlier studies of action-stopping recorded partial EMG for various purposes (*de Jong et al., 1990*; *McGarry et al., 2000*;

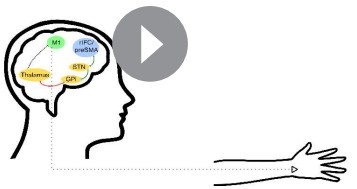

**Video 1.** Hypothetical model of the temporal cascade of processes underlying human action-stopping. Following the Go signal, after a delay, the thalamocortical drive starts building up. After a while this drive is sufficient to activate muscles via the corticospinal pathways. Following the Stop signal, the right PFC including the rIFC and the preSMA gets activated at ~ 120 ms. These region/s activate (green connections) the STN of the basal ganglia which in turn activates the globus pallidus interna which via its inhibition (red connection) on the motor regions of the thalamus cuts down the 'drive' of the motor cortex. This results in a global motor suppression at ~ 140 ms after the Stop signal. This suppression is reflected in the hand muscle at ~ 160 ms which is measured as the CancelTime. There is a delay of ~ 60 ms at the muscle level which gets added to the behavioral estimate of SSRT.

https://elifesciences.org/articles/50371#video1

*McGarry and Franks, 1997*), some more recent ones specifically interpreted the time of the partial EMG as related to stopping (*Huster et al., 2019*; *Nguyen et al., 2019*; *Raud et al., 2019*; *Raud and Huster, 2017*; *Thunberg et al., 2019*). Our results strongly affirm that partial EMG can be used to estimate the latency of stopping reflected in the muscle. Second, as just noted, they provide temporal constraints on neuroscience studies of stopping in the brain. They suggest that methods with high temporal resolution need to focus on the time after the Stop signal and before CancelTime (indeed CancelTime minus conduction time) rather than before $SSRT_{Beh}$, and the current study points to the potential of CancelTime as single-trial metric of stopping (please see our recent pre-print; *Hannah et al., 2019*). Third, our results have clinical implications. Whereas meta-analysis shows that $SSRT_{Beh}$ is longer for patients (e.g. ADHD, OCD, and substance use disorder) vs. controls (*Alderson et al., 2007*; *Bari and Robbins, 2013*; *Lavagnino et al., 2016*; *Lijffijt et al., 2005*; *Smith et al., 2014*; *Snyder et al., 2015*), not all such studies show differences (*Clark et al., 2007*; *Kalanthroff et al., 2017*; *Lipszyc and Schachar, 2010*; *Smith et al., 2014*). We predict that CancelTime will be more sensitive than $SSRT_{Beh}$. Furthermore, future studies can easily estimate within-subject variability in CancelTime, which will likely discriminate patients from controls. Fourth, our results provide insight into why $SSRT_{Beh}$ might only have a modest relationship with more 'real-world' measures of impulsivity (*Chowdhury et al., 2017*; *Enkavi et al., 2019*; *Friedman and Miyake, 2004*; *Lijffijt et al., 2004*; *McLaughlin et al., 2016*; *Skippen et al., 2019b*). As we show, $SSRT_{Beh}$ includes not only CancelTime but an extra, and variable, 60 ms ballistic stage. We expect that future studies may show stronger correlations between CancelTime and self-report than that seen between $SSRT_{Beh}$ and self-report (also see *Skippen et al., 2019b*); likewise we predict that right frontal beta burst time might also correlate more tightly with self-report measures. More generally, the detailed timing information of frontal beta at ~ 120 ms, global motor suppression at ~ 140 ms, and CancelTime at ~ 160 ms points to subprocesses of action-stopping that provide potential biomarkers that could better explain individual differences in impulse control.

In conclusion, we propose a detailed timing model of action-stopping that partitions it into subprocesses that are isolable to different nodes and are more precise than the behavioral latency of stopping. At the core of this timing model is a method of measuring the latency of stopping from the muscles. This offers a potential single-trial estimate of stopping latency that could be easily measured with minimal equipment in any lab that studies human participants.

## Materials and methods

### Participants

All were adult, healthy, human volunteers provided written informed consent and were compensated at $20/hour. The studies were approved by the UCSD Institutional Review Board (protocol #171285).

*Study 1.* Ten participants (four females; age 22 ± 1 years; all right-handed).

*Study 2.* Thirty-six participants (19 females; age 19 ± 0.4 years; all right-handed). Two were excluded for bad behavior (violating the assumptions of the independent race model - Failed Stop RT$_{Beh}$ < Correct Go RT$_{Beh}$, and P(Stop) increasing monotonically as a function of SSD), and two were excluded for noisy EMG data.

*Study 3 (TMS)*: Eighteen participants (11 females; age 19 ± 0.4 years; 15 right-handed, 2 left-handed) with no contraindications to TMS (*Rossi et al., 2011*). One was excluded for bad behavior.

*Study 4 (EEG).* Eleven participants (six females, age 19 ± 0.4 years, all right-handed).

*Study 5 (EEG)*: Fifteen participants (nine females, age 21 ± 0.4 years, all right-handed). Two were excluded from analysis, one for misaligned EEG markers due to a technical issue, while the other lacked a right frontal brain IC, based on our standard method (*Castiglione et al., 2019*; *Wagner et al., 2018*).

## Stop-signal task

This was run with MATLAB 2014b (Mathworks, USA) and Psychtoolbox (*Brainard, 1997*). Each trial began with a white square appearing at the center of the screen for 500 ± 50 ms. Then a right or left white arrow appeared at the center. When the left arrow appeared, participants had to press a key on a vertically oriented keypad using their index finger, while for a right arrow they had to press down on a key on a horizontally oriented keypad with their pinky finger (*Figure 1b inset*), as fast and as accurately as possible (Go trials). The stimuli remained on the screen for 1 s. If participants did not respond within this time, the trial aborted, and 'Too Slow' was presented. On 25% of the trials, the arrow turned red after a Stop Signal Delay (SSD), and participants tried to stop the response (Stop trials). The SSD was adjusted using two independent staircases (for right and left directions), where the SSD increased and decreased by 50 ms following a Successful Stop and Failed Stop respectively. Each trial was followed by an inter-trial interval (ITI) and the entire duration of each trial including the ITI was 2.5 s (*Figure 1a*).

*Study 1 and 2.* Participants performed the task with their right hand. They performed 40 practice trials before the actual experiment, where their baseline SSD was determined and was subsequently used as the starting SSD in the main experiment. In study 1 and 2, the experiment had 600 trials divided in 15 blocks, such that each block had 40 trials (450 Go trial and 150 Stop trials). At the end of each block the participants were presented a figure showing their mean reaction times (RT) in each block. Participants were verbally encouraged to maintain their mean reaction time constant across the different blocks and between 0.4–0.6 s.

Study 3. Participants performed the task with their left hand. Following 48 practice trials without TMS, participants performed 12 blocks of the experiment with TMS, with each block consisting of 96 trials each (72 Go trials and 24 Stop trials).

*Study 4.* Participants performed the task with their right hand. Following 160 practice trials, participants performed 4 blocks of 80 trials (240 Go trials and 80 Stop trials).

*Study 5.* Participants performed the task with their right hand. Following 80 practice trials, participants 24 blocks of 80 trials each (1440 Go trials and 480 Stop trials).

## Data recording

### EMG

EMG data were acquired using a Grass QP511 AC amplifier (Glass Technologies, West Warwick, RI) with a frequency cut-off between 30 and 1000 Hz. A CED Micro 1401 mk II acquisition system sampled the data at 2 kHz. The EMG data were acquired by CED Signal v4 software (Cambridge Electronic Design Limited, Cambridge, UK) for 2 s following the fixation cue. The data acquisition was triggered from MATLAB using a USB-1208FS DAQ card (Measuring Computing, Norton, MA). In all five experiments, surface EMG was recorded from both the first dorsal interossei (FDI) and the abductor digiti minimi (ADM) muscles of the hand (*Figure 1b inset*). In the TMS experiment, surface EMG was also recorded from the task-irrelevant right extensor carpi radialis (ECR) muscle (*Figure 4a*).

### TMS

MEPs were evoked using a TMS device (PowerMag Lab 100, MAG and More GMBH, Munich, Germany) delivering full sine wave pulses, and connected to a figure-of-eight coil (70 mm diameter,

Double coil PMD70-pCool; MAG and More GMBH, Munich, Germany). During the task, the coil was positioned on the scalp over the left primary motor cortex representation of the ECR muscle and oriented so that the coil handle was approximately perpendicular to the central sulcus, that is at ~45° to the mid-sagittal line, and the initial phase of current induced in the brain was posterior-to-anterior across the central sulcus. Prior to the experiment, the motor hot spot was determined as the position on the scalp where slightly supra-threshold stimuli produced the largest and most consistent MEPs in ECR. The position was marked on a cap worn by the participants. Resting motor threshold (RMT) was defined as the lowest intensity to evoke an MEP of at least 0.05 mV in 5 of 10 consecutive trials while participants were at rest. We then established the test stimulus intensity to be used during task, which was set to produce a mean MEP amplitude of approximately 0.2–0.5 mV whilst the participant was at rest.

MEPs were also evoked in the left FDI muscle prior to beginning the main experiment for the purpose of recording the corticospinal conduction time. The motor hot spot for the FDI was defined in a manner similar to that for the ECR. The active motor threshold (AMT) was defined as the lowest intensity to evoke a discernible MEP in 5 of 10 consecutive trials, while participants maintained slight voluntary contraction (~10% of maximum voluntary EMG amplitude during isometric finger abduction). Then, 10 stimuli were delivered at 150% AMT during slight voluntary contraction (again 10% of maximum), with the coil oriented to induce lateral-medial current in the brain in order to obtain estimates of corticospinal conduction time.

During the task, TMS stimuli were delivered on every Stop trial and on 50% of Go trials. On every Stop trial, a single TMS stimulus at the test stimulus intensity was delivered at one of six time points: inter-trial interval (100 ms prior to fixation; ITI), 100 ms, 120 ms, 140 ms, 160 ms and 180 ms after the Stop signal (*Figure 4a*). On the Go trials, TMS stimuli were yoked to the time of the Stop signal on the previous Stop trial. Thus, there were 48 trials per TMS time point on Stop trials and 96 trials per time point on Go trials.

## EEG
64 channel EEG (Easycap, Brainvision LLC) was recorded in the standard 10/20 configuration at 1 kHz using BrainVision actiChamps amplifier (Brain Products GMBH, Gilching, Germany) and BrainVision Pycorder (Brain Products GMBH, Germany).

## Data analysis
All analyses were performed using MATLAB (R2016b, R2018b, R2019a).

### Stop signal reaction time
SSRT from the behavioral responses ($SSRT_{Beh}$) was determined using the integration method (*Verbruggen et al., 2019*). When calculating SSRT using the EMG responses, $SSRT_{EMG}$, as the P(Respond|Stop) was often much more than 0.5, we calculated the SSRT individually for all SSDs and then averaged it (*Verbruggen and Logan, 2009*).

### EMG data analysis
EMG data were filtered using a 4[th] order Butterworth filter (roll-off 24 dB/octave) to remove 60 Hz noise and its harmonics at 120, and 180 Hz. EMG data were full-wave rectified and the root-mean square (RMS) of the signal was computed using a centered window of 50 ms. Any EMG activity which was greater than 8 SD of the mean EMG activity in the baseline period (Fixation to Go cue) was marked, on a trial-by-trial basis. Starting from the peak of that EMG activity, we backtracked and marked the onset at the point where the activity dropped below 20% of the peak for five consecutive ms. This method of adjusting the threshold based on the peak EMG activity, allowed better onset detection than a fixed threshold, especially when the amplitude of the EMG activity was small. The time when EMG started to decline was determined as the time when, following the peak EMG, the activity decreased for five consecutive ms. Visual inspection of individual trials showed that this method provided a reliable detection of both EMG onsets (see *Figure 1—figure supplement 1a,b* for $RT_{EMG}$ vs. $RT_{Beh}$ correlation) and decline. Any detected EMG timing which was beyond 1.5 times the inter-quartile range (IQR) of the first and third quartile (Q3) of that particular timing distribution was deemed an outlier. This removed < 4% trials. CancelTime was marked as the time of the

EMG decline following the Stop signal. For outlier rejection, CancelTimes had a lower cutoff of 50 ms and higher cutoff of Q3+1.5 × IQR. This removed < 3% trials.

As the peak EMG amplitude for the FDI and ADM muscle were quite distinct, before averaging the two EMG activities, we normalized the muscle activity by the peak activity in that particular muscle (Voltage$_{Norm}$ in *Figures 2a, b* and *3c*, *Figure 3—figure supplement 2*).

## Global MEP suppression

MEP amplitudes were measured on a trial-by-trial basis. Data were included for analysis if the following criteria were met: (i) the amplitude of the ECR EMG signal in a 90 ms period prior to the TMS stimulus was < 0.05 mV; (ii) the amplitude of the MEP fell within the mean±1.5× IQR of values for the same time point and trial type (Correct Go, Failed Stop, Successful Stop). Thereafter, MEP amplitudes measured at the ITI were collapsed across trial type (Correct Go, Failed Stop and Successful Stop), averaged and used as a baseline against which to compare other TMS time points. For each of the other TMS time points (100, 120, 140, 160, 180 ms following the Stop signal), data were averaged within each trial type (Correct Go, Failed stop, Successful Stop) and expressed as a percentage of the mean ITI MEP amplitude.

## Corticospinal conduction time

Corticospinal conduction time was determined by delivering TMS over the hand representation of left FDI and measuring MEP from the muscle (*Figure 4c*). The earliest MEP onset latency across 10 trials was identified by visual inspection of the EMG traces (*Hamada et al., 2013*; *Hannah and Rothwell, 2017*; *Rossini et al., 2015*).

## Trial-by-trial analysis of CancelTime and time of global motor suppression

To compare the temporal association between the EMG decline and MEP suppression, we performed a trial-by-trial analysis of Stop-signal task data only on trials where an EMG burst was detected. We first normalized the time of TMS on a given trial by subtracting the time of EMG decline from the time of the TMS pulse. Hence, negative values mean that TMS was delivered before the EMG decline and positive values mean that TMS was delivered after. We then plotted MEP amplitudes for each of the three response types (Correct Go, Failed Stop, and Successful Stop) against the normalized times binned into 30 ms windows. This analysis meant that for a given individual there were relatively few trials per time bin, and some bins would occasionally contain no data. Therefore, we combined data across all individuals. Prior to this, MEP amplitudes for each individual were normalized to the mean MEP amplitude at the inter-trial interval, to account for inter-individual variability in absolute MEP amplitudes at baseline. We restricted our analysis to time bins that contained at least 50 trials, which resulted in time range −90 ms to 60 ms.

## EEG preprocessing

We used EEGLAB (*Delorme and Makeig, 2004*) and custom-made scripts to analyze the data. The data were downsampled to 512 Hz and band-pass filtered between 2–100 Hz. A 60 Hz and 180 Hz FIR notch filter were applied to remove line noise and its harmonics. EEG data were then re-referenced to the average. The continuous data were visually inspected to remove bad channels and noisy stretches.

## ICA analysis

The noise-rejected data were then subjected to logistic Infomax ICA to isolate independent components (ICs) for each participant separately (*Bell and Sejnowski, 1995*). We then computed the best-fitting single equivalent dipole matched to the scalp projection for each IC using the DIPFIT toolbox in EEGLAB (*Delorme and Makeig, 2004*; *Oostenveld and Oostendorp, 2002*). ICs representing non-brain activity related to eye movements, muscle, and other sources were first identified using the frequency spectrum (increased power at high frequencies), scalp maps (activity outside the brain) and the residual variance of the dipole (greater than 15%) and then, subtracted from the data. A putative right frontal IC was then identified from the scalp maps (if not present then we used frontal topography) and the channel data were projected onto the corresponding right frontal IC. The data on Successful Stop trials were then epoched from −1.5 s to 1.5 s aligned to the Stop signal. We

estimated the time-frequency maps from 4 to 30 Hz, and −100 to 400 ms using Morlet wavelets with three cycles at low frequencies linearly increasing by 0.5 at higher frequencies. The IC was selected only if there was a beta power (13 to 30 Hz) increase in the window between the Stop signal and SSRT_{Beh} compared to a time-window *prior to the Go* cue (−1000 to −500 ms aligned to Stop signal). In each participant, the beta frequency which had the maximum power in this time window was used in the beta bursts computation (*Figure 5—figure supplement 1*).

### Beta bursts

To estimate the beta bursts, the epoched data were first filtered at the peak beta frequency using a frequency domain Gaussian window with full-width half-maximum of 5 Hz. The complex analytic envelope was then obtained by Hilbert transform, and its absolute value provided the power estimate. In each participant, to define the burst threshold, the beta amplitude within a period of 500 to 1000 ms (*i.e.* after the Stop signal in the Stop trials, and after the mean SSD in the Correct Go trials) was pooled across all trials [Note that compared to the ICA analysis here we picked a different time-window to estimate the burst threshold to keep the analysis unbiased. However, picking the same time-window also yielded similar results]. The threshold was set as the median + 1.5 SD of the beta amplitude distribution (*Figure 5—figure supplement 3*). Once the burst was detected, the burst width threshold was set as the median + 1 SD. We binary-coded each time point where the beta amplitude crossed the burst width threshold to compute the burst % across trials. For each detected burst, the time of the peak beta amplitude was marked as the BurstTime.

## Statistical analysis

For pairwise comparisons, the data were first checked for normality using Lilliefors test, and if normally distributed a two-tailed *t*-test (*t*-statistic) was performed, else a Wilcoxon signed rank test (*Z*-statistic) was performed. We interpret the effect sizes as small (Cohen's *d*: 0.2–0.5; Bayes Factor in favor of the alternate hypothesis, $BF_{10}$: 1–3), medium (*d*: 0.5–0.8; $BF_{10}$: 3–10), large (*d* > 0.8; $BF_{10}$ > 10). For comparisons across multiple levels, repeated-measures ANOVA was used, followed by Bonferroni corrected *t*-tests for pairwise comparisons (Bonferroni corrected *p*-value: $p_{Bon}$). The Greenhouse-Geisser correction was applied where the assumption of sphericity in ANOVA was violated (corrected *F*-statistic: $F_{GG}$). Effect sizes for ANOVAs were interpreted as small (partial eta-squared, $\eta_P^2$: 0.01–0.06), medium ($\eta_P^2$: 0.06–0.14), and large ($\eta_P^2$ : 0.14). For correlational analyses, Pearson's correlation coefficient (*r*) was usually used, but Spearman's correlation coefficient (*ρ*) was used when the data was bounded in a closed interval. All data are presented as mean ± s.e.m.

In testing the relationship between BurstTime and CancelTime, we performed a permutation test. We sampled BurstTimes randomly from a uniform distribution between 0 and SSRT_{Beh} for a given participant for 3000 iterations. For each iteration, we then computed the correlation (*r*) between the mean BurstTime and the mean CancelTime across participants. This generated a distribution of *r* ranging between −1 and 1. The *p*-value for our analysis was determined as the $P(r \geq r_{Obs}|H_0)$ in the permuted data.

## Bayesian modelling of behavioral data

We used the BEESTS model developed by Dora Matzke and colleagues (run in R Studio 1.1.463) which assumes a race between two stochastically independent process, a Go and a Stop processes. This model estimates the distribution of the SSRT by using the participant's Go RT_{Beh} distribution, and by considering the Failed Stop RT_{Beh} as a censored Go RT_{Beh} distribution. The censoring points are sampled randomly from the SSRT distribution on each Stop trial. The RT_{Beh} distributions underlying the Go and Stop processes are assumed to have a Gaussian and an exponential component and is described by three parameters ($\mu_{Go}$, $\sigma_{Go}$, $\tau_{Go}$ and $\mu_{Stop}$, $\sigma_{Stop}$, $\tau_{Stop}$). For such ex-Gaussian distributions, the mean and variance of the RT_{Beh} distributions are determined as $\mu + \tau$ and $\mu^2 + \tau^2$, respectively. The model also estimates the probability of trigger failures for each participant. The model uses Bayesian Parametric Method (BPE) to estimate the parameters of the distributions. We used a hierarchical BPE, where individual subject parameters are modeled with the group-level distributions. This approach is thought to be more accurate than fitting individual participants and is effective when there is less data per participant (*Matzke et al., 2013*). We pooled the subjects across both study 1 and 2 to estimate the individual parameters. The priors were bounded uniform

distributions ($\mu_{Go}$, $\mu_{Stop}$: $U(0,2)$; $\sigma_{Go}$, $\sigma_{Stop}$: $U(0,0.5)$ $\tau_{Go}$, $\tau_{Stop}$: $U(0,0.5)$; pTF: $U(0,1)$). The posterior distributions were estimated using the Metropolis-within-Gibbs sampling and we ran multiple chains. We ran the model for 5000 samples with a thinning of 5. The Gelman-Rubin ($\hat{R}$) statistic was used to estimate the convergence of the chain. Chains were considered converged if $\hat{R}$<1.1.

### Data and scripts

A core element of this paper is a novel method of calculating single-trial stopping latency from EMG. Accordingly, we provide the EMG and behavioral data from all participants in study 1 and 2, along with analysis scripts, and a brief description of how to execute the scripts (https://osf.io/b2ng5/). All other EMG, TMS-MEP and EEG data and scripts are also provided at the above link.

## Acknowledgements

We thank Dora Matkze for sharing the scripts for BEESTS modelling, Sven Bestmann for insightful comments on data, Simon Little for sharing the beta-burst analysis script, Kelsey Sundby for sharing some EEG and EMG data, and Xinze Yu and Hunter Robbins for help in data recording. We gratefully acknowledge our support from NIH: NS106822 and DA026452.

## Additional information

### Competing interests

Adam R Aron: Reviewing editor, *eLife*. The other authors declare that no competing interests exist.

### Funding

| Funder | Grant reference number | Author |
| --- | --- | --- |
| National Institutes of Health | NS 106822 | Sumitash Jana<br>Ricci Hannah<br>Vignesh Muralidharan<br>Adam R Aron |
| National Institutes of Health | DA 026452 | Sumitash Jana<br>Ricci Hannah<br>Vignesh Muralidharan<br>Adam R Aron |

The funders had no role in study design, data collection and interpretation, or the decision to submit the work for publication.

### Author contributions

Sumitash Jana, Ricci Hannah, Vignesh Muralidharan, Conceptualization, Data curation, Software, Formal analysis, Validation, Investigation, Visualization, Methodology, Writing - original draft, Writing - review and editing; Adam R Aron, Conceptualization, Resources, Formal analysis, Supervision, Funding acquisition, Validation, Visualization, Writing - original draft, Project administration, Writing - review and editing

### Author ORCIDs

Sumitash Jana (iD) https://orcid.org/0000-0003-3742-3958
Ricci Hannah (iD) https://orcid.org/0000-0001-5379-3292

### Ethics

Human subjects: All human volunteers provided written informed consent prior to their participation. The participants were compensated at $20/hour. The University of California San Diego Institutional Review Board approved all the studies (protocol #171285).

Decision letter and Author response
Decision letter https://doi.org/10.7554/eLife.50371.sa1
Author response https://doi.org/10.7554/eLife.50371.sa2

## Additional files

### Supplementary files

• Transparent reporting form

### Data availability

A core element of this paper is a novel method of calculating single-trial stopping speed from EMG. Accordingly, we provide the EMG and behavioral data from 10 participants in study 1, along with analysis scripts, and a brief description of how to execute the scripts (https://osf.io/b2ng5/).

The following dataset was generated:

| Author(s) | Year | Dataset title | Dataset URL | Database and Identifier |
|---|---|---|---|---|
| Jana S, Hannah R, Muralidharan V, Aron A | 2019 | Temporal cascade of frontal, motor and muscle processes underlying human action-stopping | https://osf.io/b2ng5/ | Open Science Framework, b2ng5 |

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

# Appendix 1

The difference between $SSRT_{Beh}$ and the time of EMG cancellation (CancelTime) might be attributed to an inherent 'ballistic stage' in movements, and once the muscle activity crosses the point-of-no-return they can no longer be stopped, making the movement inevitable (*de Jong et al., 1990*; *Mirabella et al., 2006*; *Osman et al., 1986*; *Verbruggen and Logan, 2009*). The duration of such ballistic stages has been estimated to be 10–25 ms in saccades in non-human primates (*Boucher et al., 2007*; *Kornylo et al., 2003*; *Purcell et al., 2010*) and 40–50 ms for reaching movements in humans (*Gopal and Murthy, 2016*; *Jana and Murthy, 2018*). In other words, the time of EMG cancellation on partial EMG trials reflects a time just before the point-of-no-return, whereby if EMG activity is allowed to develop beyond this point it will exceed a critical threshold such that a button press necessarily ensues (we presume this threshold reflects the point at which the inertia of the finger is overcome). In this respect, what is being tracked by the SSD staircasing procedure is the probability of crossing that EMG threshold, but since SSRT is calculated based on keypress response times, it inevitably incorporates the ballistic stage that follows the crossing of this threshold.

To test the idea that there might be a ballistic stage in responses which inflates $SSRT_{Beh}$ we simulated the independent race model with a Go and Stop accumulator which raced to a threshold, where the activity of each accumulator could be described by the mean drift rate ($\mu$) and the SD of the drift rate ($\sigma$) (*Boucher et al., 2007*; *Ramakrishnan et al., 2012*; *Usher and McClelland, 2001*). First, we tested whether the independent race model without a ballistic stage could fit the behavioral results. For each participant, we fitted the $RT_{EMG}$ and $RT_{Beh}$ distributions in the Correct Go trials, to estimate the ($\mu_{GO-EMG}$, $\sigma_{GO-EMG}$) and ($\mu_{GO-Beh}$, $\sigma_{GO-Beh}$). Next, using the estimated Go parameters, we estimated the ($\mu_{Stop}$, $\sigma_{Stop}$), which would best fit the inhibition function. This estimation of the Stop parameters ($\mu_{Stop}$, $\sigma_{Stop}$) was performed separately for EMG ($\mu_{STOP-EMG}$, $\sigma_{STOP-EMG}$) and keypresses ($\mu_{STOP-Beh}$, $\sigma_{STOP-Beh}$). We reasoned that if the only difference between the two inhibition functions is due to the difference in the Go processes ($RT_{EMG}$ vs $RT_{Beh}$) then the $\mu_{STOP-EMG}$ should be similar to $\mu_{STOP-Beh}$. However, this was not the case. First, Stop distributions estimated from the EMG and keypresses were quite distinct, where $\mu_{STOP-EMG}$ (188 ± 8 ms) was significantly less than $\mu_{STOP-Beh}$ (207 ± 5 ms; $t(41)$ = 2.7, p=0.009, $d$ = 0.4, $BF_{10}$ = 4.4). Second, the difference between the simulated and experimental inhibition function was much greater when ($\mu_{STOP-EMG}$, $\sigma_{STOP-EMG}$) was used to fit the behavioral inhibition function compared to the EMG inhibition function (*Figure 3—figure supplement 1a,b*). To quantify this, we computed the squared difference between the experimental and simulated P(Respond|Stop) data (*Figure 3—figure supplement 1c*; EMG inhibition function, squared error between experimental and simulated data = 0.15 ± 0.02; RT inhibition function, squared error = 0.23 ± 0.02, $t(41)$ = 2.7, p=0.011, $d$ = 0.6, $BF_{10}$ = 3.7). Conversely, the difference between the simulated and experimental inhibition function was much greater when $\mu_{STOP-Beh}$ was used to fit the EMG inhibition function compared to the behavioral inhibition function (*Figure 3—figure supplement 1d,e,f*; behavioral inhibition function, squared error = 0.10 ± 0.01; EMG inhibition function, squared error = 0.22 ± 0.02, $t(41)$ = 4.7, p<0.001, $d$ = 0.7, $BF_{10}$ >100). This incompatibility suggests that some change needs to be made to the model such that one Stop distribution is able to fit both the EMG and keypress inhibition functions. Hence, we tested whether incorporating a ballistic stage in the delay between $RT_{EMG}$ and $RT_{Beh}$ could rescue the model.

To check this, we used a model previously used to study stopping of reaching movements in the context of coordinated eye-hand movements (*Gopal and Murthy, 2016*; *Jana and Murthy, 2018*). Here, the Go process comprises an accumulation phase, and once the accumulator hits the threshold, EMG responses can be observed, and then following a peripheral delay, the EMG builds up enough (i.e. the point-of-no-return) to be able to cross

the inertia of the limb and generate a movement (see *Figure 3—figure supplement 1g,h,i*). If the Stop reaches the threshold before the Go reaches the threshold, then no EMG is elicited (no EMG trial). Further, if the Stop reaches the threshold early during the delay period when the EMG has not built up to a critical level (point-of-no-return) it will be able to inhibit the response resulting in a partial EMG trial. However, if the Stop reaches the threshold after the EMG has reached the point-of-no-return a movement is inevitable, resulting in a Failed Stop trial. We used this model to estimate the duration of the ballistic stage. Across participants, the mean ballistic stage was $34 \pm 4$ ms. Incorporation of this ballistic stage allowed $\mu_{STOP\text{-}EMG}$ to fit the behavioral inhibition function much better than a model without a ballistic stage (*Figure 3—figure supplement 1j,k*; squared error = $0.14 \pm 0.02$, $t(41) = 4.2$, p<0.001, $d = 0.7$, $BF_{10} > 100$). Thus, based on our model, there exists a ballistic stage in keypress responses, which might be responsible for the difference between CancelTime and $SSRT_{Beh}$. While this measure was less than our estimate of ~60 ms, we note that some participants did not have a sigmoidal inhibition function (as the task was designed to have a P(Stop)=0.5 such that $SSRT_{Beh}$ could be well estimated) leading to suboptimal estimation of the Stop parameters. Indeed, when we considered only those participants who had lower than the population median error, the duration of the ballistic stage was $47 \pm 3$ ms which is closer to our estimate of ~60 ms.

## Methods

The rate of accumulation is governed by the differential equations (*Boucher et al., 2007*; *Ramakrishnan et al., 2012*; *Usher and McClelland, 2001*):

$$da_{GO} = \frac{dt}{\tau}\left[\mu_{GO} - k.a_{GO}(t)\right] + \sqrt{\frac{dt}{\tau}}\xi_{GO}$$

$$da_{STOP} = \frac{dt}{\tau}\left[\mu_{STOP} - k.a_{STOP}(t)\right] + \sqrt{\frac{dt}{\tau}}\xi_{STOP}$$

where $da_{GO}$ and, $da_{STOP}$ represents the change in accumulation within a time-step $dt$ and a time constant $\tau$; $\mu_{GO}$ and $\mu_{STOP}$ is the mean drift rate; $\xi_{GO}$ and $\xi_{STOP}$ is a Gaussian noise term with mean 0 and SD equal to the $\sigma$ of the associated accumulator, which represents the noise in the input signal; $k$ is the leakage parameter. $k$ was set to 0, $\frac{dt}{\tau}$ was set to 1.

## Parameter estimation of the independent race model with a ballistic stage

**Step 1** was to estimate the ($\mu_{GO\text{-}EMG}$, $\sigma_{GO\text{-}EMG}$), that is the mean and SD that would generate $RT_{EMG}$ distribution (*Figure 3c inset* green beeswarm plot) for each participant. A range of values, representing the parameter space for ($\mu_{GO\text{-}EMG}$, $\sigma_{GO\text{-}EMG}$), which could generate behaviorally relevant distributions were uniformly and 'coarsely' sampled, to simulate distributions for 2000 trials. The top 20 parameters which generated a distribution with a mean and SD within 30 ms of the mean and SD of the empirical $RT_{EMG}$ distribution and had the minimum least-squared error between the empirical and simulated cumulative distribution function, were fed into MATLAB's fmincon function for optimization. The MATLAB function tried to minimize the least-squared difference between the empirical and simulated cumulative distribution functions (this method of coarse sampling followed by fmincon minimization of the top 20 parameters was used for all the subsequent steps). Additionally, for the optimization, a nonlinear constraint was imposed such that the absolute difference between the mean of the empirical and simulated RT distribution was < 10 ms.

For each participant we were able to estimate the ($\mu_{GO\text{-}EMG}$, $\sigma_{GO\text{-}EMG}$) well, such that the simulated distribution had a mean similar to the empirical one. Within each participant, the simulated $RT_{EMG}$ closely matched the mean of the simulated $RT_{EMG}$ (2-tailed unpaired $t$-test: all p>0.05; $BF_{10}$: min = 0.06, max = 0.85, mean = $0.08 \pm 0.01$). Across all participants, the

estimated EMG onset distribution closely matched the empirical mean (Empirical: 343 ± 12 ms, Simulated: 343 ± 13 ms, $t(41)$ = 0.6, p=0.532, $d$ = 0.1, $BF_{10}$ = 0.2).

**Step 2** was to estimate a delay which could be added to the $RT_{EMG}$ which would result in the $RT_{Beh}$ distribution. In other words, the sum of the Go-EMG accumulation process described by ($\mu_{GO\text{-}EMG}$, $\sigma_{GO\text{-}EMG}$) and the delay would yield the $RT_{Beh}$ distribution. A range between 10–300 ms was coarsely sampled, and then the top 20 parameters were fed into fmincon for optimization. Additionally, a nonlinear constraint was imposed such that the absolute difference between the mean of the empirical and simulated data was < 10 ms.

At single participant level, the estimated ($\mu_{Delay}$, $\sigma_{Delay}$) yielded distributions that closely matched the mean (of the empirical $RT_{Beh}$ distribution (2-tailed unpaired $t$-test: all p>0.05; $BF_{10}$: min = 0.06, max = 0.13, mean = 0.074 ± 0.003). Thus, we were able to estimate all the parameters ($\mu_{GO}$, $\sigma_{GO}$, and $\mu_{Delay}$, $\sigma_{Delay}$) describing the putative Go process. Across the population, the simulated RT distributions closely matched the empirical mean (Empirical: 478 ± 12 ms, Simulated: 478 ± 13 ms, $t(41)$ = 0.04, p=0.967, $d$ = 0.01, $BF_{10}$ = 0.2). Further, across the population, the simulated delay period (133 ± 3 ms) was not significantly different than the empirical delay between the $RT_{EMG}$ and $RT_{Beh}$ (135 ± 3 ms; $t(41)$ = 1.9, p=0.068, $d$ = 0.3, $BF_{10}$ = 0.8).

**Step 3** was to estimate the ($\mu_{Stop}$, $\sigma_{Stop}$) from the EMG responses using the independent race model (we assumed that the true Stop distribution could be estimated from the EMG inhibition function). Thus, the end of the Go-EMG process marked the onset of the EMG response (described by the estimated ($\mu_{GO\text{-}EMG}$, $\sigma_{GO\text{-}EMG}$)). The EMG inhibition function was calculated based on whether EMG was detected (thus partial EMG responses were considered as error responses), and was fitted with a cumulative Weibull function (**Hanes et al., 1998**; **Ramakrishnan et al., 2012**):

$$w(t) = \gamma - (\gamma - \delta)e^{[-(t/\alpha)^{\beta}]}$$

(where $t$ is the SSD, $\alpha$ is the time at which the function reaches 64% of its full growth, $\beta$ is the slope, $\delta$ is the minimum value of the function, and $\gamma$ is maximum value of the function. The difference between $\delta$ and $\gamma$ marks the range of the inhibition function). The optimization process tried to minimize the least-squared error between the fits of the empirical and simulated inhibition functions. Additionally, a nonlinear constraint was imposed that the least-squared error between the empirical and simulated data was < 0.2.

**Step 4** was to estimate the duration of the ballistic stage. Here the duration of the Go process (keypress response time) would be the sum of the ($\mu_{GO\text{-}EMG}$, $\sigma_{GO\text{-}EMG}$) accumulation and the delay ($\mu_{Delay}$, $\sigma_{Delay}$). The Stop process was the same as that estimated from EMG, and we used these parameters to simulate a race model that would best fit the keypress inhibition function. The ballistic stage was assumed to have a normal distribution and again the least-squared error between the fits of the empirical and simulated inhibition functions was minimized. A range between 5–200 ms was coarsely sampled, and then the top 20 parameters were fed into fmincon for optimization. Additionally, a nonlinear constraint was imposed that the least-squared error between the empirical and simulated data was < 0.2.

To simulate the independent race model without a ballistic stage (i.e. when considering $RT_{Beh}$ as the Go process, and directly fitting the RT inhibition function to estimate the underlying Stop distribution), we followed Step 1 to estimate the Go parameters, and Step 3 to estimate the Stop parameters.

