## [Decision Letter]

Thank you for submitting your article "Temporal cascade of frontal, motor and muscle processes underlying human action-stopping" for consideration by *eLife*. Your article has been reviewed by three peer reviewers, including Wery van den Wildenbert as the Reviewing Editor and the evaluation has been overseen by Richard Ivry (Senior Editor).

The Reviewing Editor, Wery van den Wildenberg, drafted this letter based on the detailed evaluation reports. As you will see, the reviewers are generally positive about the broad scope and they commend the use of different methodological approaches.

Several issues, detailed below as major points, deserve attention when preparing a revision of your manuscript. Of these, we would like to highlight the tendency to not give sufficient credit to prior relevant publications of others (some of which are listed under major point 1). The inclusion of relevant previous work on partial EMG and beta-burst analyses (major point 10) might lead to a more balanced rating of the acclaimed novelty of the methods and conclusions.

Given that this paper addresses questions about when and where stopping occurs, major points 6 and 11 seem to be particularly fundamental. Major point 6 might be viewed as a fundamental challenge to the temporal cascade as proposed by the authors (i.e., about when stopping occurs). Major point 11 seems to fundamentally challenge the attempt to spatially localize a stop command that is necessary and sufficient for stopping (i.e., about where stopping occurs).

The work is presented as if it "confirms a detailed model of action-stopping" [Abstract]. In general, the reviewers seem to be in agreement in their impression that this should better be toned down (see also major point 4). In its current form, which in and of itself represents elaborate work, such claims might better be framed in terms of tentative evidence for a neural model for stopping.

Summary:

The in-depth content and broad scope of the manuscript are impressive. The five empirical studies cover different neurocognitive levels of the information processing chain when healthy participants are cued to stop a motor action. Not only is stopping investigated at different stages during processing (from the brain to muscles, and behavior), the lineup of the methods, including EEG, single-pulse TMS, EMG, and modeling techniques, provide a full account of the temporal dynamics and the neural mechanisms that are at play during action stopping. The fact that the extent of these comments go far beyond the recommendation of 500 words reflects the width of methods and topics touched by this manuscript. Whereas some interpretations and claims should be toned down a bit, and the bulk of the comments refer to these rather conceptual issues, the work is considered very important and an innovative contribution to the study of action cancellation.

Essential revisions:

Relevant Background Literature:

1) The authors are kind to mention work by Huster et al. on residual or partial EMG in successful stop trials. It should be noted though that earlier work already went into a similar direction yet does not seem to receive enough credit (in general, and not specifically with respect to this manuscript). The authors may thus consider including references to this earlier work. Some examples:

- de Jong et al., 1990.

- McGarry (1999). On the nature of stopping a voluntary action. Dissertation, University of British Columbia, Canada. (see also McGarry's articles from 2003 in Motor Control and Q J Exp Psychol).

- Goonetilleke et al. (2010). A within-trial measure of the stop signal reaction time in a head-unrestrained oculomotor countermanding task. J Neurophysiol.

- And last but not least, some more recent articles/manuscripts:

- Atsma et al. (2018). Active Braking of Whole-Arm Reaching Movements Provides Single-Trial Neuromuscular Measures of Movement Cancellation. J Neurosci.

- Raud et al. (submitted). Differences in unity: the go/no-go and stop signal tasks rely on different inhibitory mechanisms. bioRxiv 705079; doi: https://doi.org/10.1101/705079

- Huster et al. (submitted). The P300 as marker of inhibitory control – fact or fiction? bioRxiv 694216; doi: https://doi.org/10.1101/694216

- Thunberg et al. (submitted). Stimulating stopping? Investigating the effects of tDCS over the inferior frontal gyri and visual cortices. bioRxiv 723296; doi: https://doi.org/10.1101/723296

- In light of this relatively rich prior history, the authors may want to consider to tone down their claim on this approach (E.g., "We predict that our new single-trial method of CancelTime will be more sensitive than SSRTBeh.").

SSRTEMG measure of Stopping:

2) I am admittedly torn when it comes to the SSRTEMG. On the one hand, one may argue that it could avoid rather unspecific biases associated with the use of different response systems for the estimation of the SSRT. Then again, whereas plausible, it is not really clear that this is the case as little research systematically assessed the influence of response devices on the shape and timing of the EMG. Also, I see no good justification for classifying partial response EMG trials as failed stop trials; no overt response was produced, which is the instruction given to the participants, and I think there is no good justification to only classify trials without any EMG response as successful stop trials. In the end, I am unsure about how to interpret this measure meaningfully, and whether it can be used to "validate" the CancelTime (as is implied here). Not least, the calculation of the SSRTEMG relies on a rather small subset of SSDs (three per subject), as described in the subsection “Data analysis”, whereas usually a wide sampling of SSDs is recommended.

3) When testing the partial response EMG against different SSDs, the H0 of no difference is justified by means of "weak go processes". I am wondering though what that implies: a very slow response (i.e., shallow slope), or one with a high threshold (thus unlikely to cause a button press), or a relatively delayed onset?

Functional interpretation of CancelTime:

4) The authors stress repeatedly that the CancelTime can be considered or is used as single-trial measure of the stop process. I suggest to tone this claim down, because:

- it is not really used as a single-trial measure in the vast majority of analyses (with exceptions being the SD-based correlations of CancelTime and SSRTBEESTS, or the CancelTime-MEP alignment).

- a proper metric should undergo testing in terms of its psychometric properties (reliability, validity, objectivity etc.). This is too often overlooked, not only but especially with psychophysiological measures.

Thus, whereas I am also hopeful that the CancelTime may serve us as such a measure one day, I think we should not make this claim yet.

5) In order to validate CancelTime, the authors show high correlations with other stop task measures including SSRTbeh and BEESTs model estimates. Along with other traditional measures, CancelTime was positively correlated with trigger failure rate. This seems to challenge the discriminant validity of CancelTime (and/or trigger failure rate). CancelTime was calculated on trials in which a stop signal has been triggered, reaches the muscle of the hand, but a behavioral response is not emitted. It is also intended to be a purer measure of response inhibition. Trigger failures are putative failures of attention and should only occur on failed stop trials. I suggest that the authors discuss this relationship between CancelTime and trigger failure rate.

Concerning Temporal Dynamics:

6) If I am understanding correctly, CancelTime is computed exclusively from partial EMG trials. Additionally, it is intended to be an alternative to SSRTbeh, which is a measure of the central tendency of SSRT across all trials. When CancelTime is compared to SSRTbeh, CancelTime is ~60ms shorter. The authors argue that CancelTime is shorter than SSRTbeh because CancelTime does not include the (~60ms) electromechanical delay and therefore is a purer measure of stopping. However, I would like to suggest another explanation for why CancelTime is shorter than SSRTbeh.

Assuming (1) a race model architecture, (2) variability in the go RT and (3) variability in SSRT, then failed stops will tend to occur when SSD is long, go RT is fast, and SSRT is slow. Successful stops will tend to occur when SSD is short, go RT is slow, and SSRT is fast. Partial EMG trials, which I understand to be trials in which the go process doesn't win as decisively as a successful stop trial or fail as decisively as a failed stop trial, should have SSD, go RT, and SSRT values between these two extremes. However, ~1/2 of trials are failed stop, ~1/4 of trials as partialEMG, and ~1/4 of trials are successful stops. Therefore, partialEMG trials should be preferentially sampled from the half of trials with shorter SSD, slower go RT, and faster SSRT. If this is the case, then CancelTime being faster than SSRTbeh may be partially or completely explained by CancelTime being selectively sampled from faster SSRT trials whereas SSRTbeh is computed across all stop trials.

To concretize my suggestion in an example, if we assume an SSRT distribution with variability such that the first quantile has a mean of 120 ms, the second 160 ms, the third 250, and the fourth 350 ms. When the first quantile is sampled, these should preferentially result in successful stops, when the second is sampled these should preferentially result in partialEMG trials, and when either of the latter two are sampled these should preferentially result in failed stop. However, if you computed CancelTime on these data it should be closer to 160ms (as a result of preferentially sampling the 2nd quantile) but the overall central tendency of the SSRT distribution would be ~220ms.

This alternative implies that the average neural latency for stopping may not differ from SSRTbeh, or may differ less than the ~60 ms suggested here. One tool for evaluating this alternative explanation may be to simulate stop trials with variability in go, stop and SSD and evaluate whether computing SSRT only (or preferentially) on trials sampled from the 2nd quartile of SSRT can produce significant differences from the traditional method of computing SSRTbehav. If it may be useful, here is some openly available code that instantiates the interactive race framework and would allow varying go, stop, and SSD: https://github.com/bissettp/SharingContextDependence/blob/master/interactive_race.ipynb).

7) The authors stress that parts of their model of the processing sequence is supported by the fact that the temporal distance between cortical reduction in excitability (TMS) and the CancelTime corresponds to the corticospinal conduction time (here about 23 ms). The difference between the CancelTime and the reduced motorcortical excitability is about 15 ms (155 – 140). However, the onset of decreased motorcortical excitability, which would best correspond to the onset of the decline in MEP amplitude, does start before that, as can be seen in Figure 4A. An only somewhat smaller effect is apparent already at 120ms, with an onset probably even before that. This would add another 20+ ms to the equation, thus contrasting 23ms vs. 35+ms. The point I want to make is that, even though I find the interpretation of the findings interesting and plausible, care has to be taken not to over-interpret these measures since they are heavily dependent on our statistical procedures (sample size, size of presumably relevant effect, parameterization of onset times etc.).

8) The validity of the reduced TMS-MEPs as indicators of the stopping process is derived from the strong reduction in successful stop relative to go trials around 140 ms, a notion further supported by Figure 4D. It is not discussed, however, that MEPs also are reduced for unsuccessful stop trials around 140 ms locked to the stop signal (a reduction similar to that of successful stop trials; Figure 4B; the effect does not seem to be that much smaller). How to reconcile these findings?

9) In the fourth paragraph of the Discussion, the authors cite papers showing that in monkey, rats, and humans that neurons that are putatively related to stopping change close to SSRT. This seems like considerable evidence against the chronometric framework presented here that assumes that the stop command is sent from brain to muscle well before SSRT (~60 ms + conduction time). I think it would be helpful to understand the authors' position on how these seemingly contradictory findings relate to this current framework.

Functional Significance of Beta Burst Activity:

10) How does the work presented in Study 4 relate to the recent preprint of a paper by Wessel on the role of beta bursts in stopping? [Wessel, J.R., "β-bursts reveal the trial-to-trial dynamics of movement initiation and cancellation", https://www.biorxiv.org/content/10.1101/644682v1].

11) Study 4 finds an increase in beta bursts for successful stops relative to baseline and go trials, an effect interpreted as indicator of the engagement of the right inferior frontal stopping processes. This is a very interesting finding, but its interpretation is not without some conceptual problems.

- Given that EEG is plagued with the inverse problem, it seems somewhat unjustified to claim that this signature originates in the right frontal cortex. But even if one would agree that a rough deduction of the source from its topography is possible, I would have a hard time to associate the rather midfrontal topography with a right frontal source close to the sylvian fissure. Beyond, the authors performed dipole fitting on the independent component topographies, and thus could easily check if the individual dipoles of these components at least roughly are fitted to such a position.

- Βeta burst activity is depicted and analyzed relative to the pre-stop period. If we leave the IC-selection aside for a moment, the resulting time courses may still be in accordance with an effect akin to motor beta: around the time of the stop signal, beta activity would be similarly low for all three categories due to motor preparation, followed by rebound-effects that might be earlier in case of successful stops (cancelled), succeeded by unsuccessful stops and go trials (rebound after execution).

The IC selection states, however, that a component needed to show a beta power increase between the stop signal and the SSRT compared to a time window prior to the go cue. This should exclude potential motor beta components with typical time courses delineated above. The time course for the go activity supports this assumption. Nevertheless, given the conceptual importance of this beta component in this context, it would be nice to also see the go-locked time courses for the stop-related conditions.

12) I understand the presented beta burst data to be the probability that a beta burst occurred on any given trial. Assuming this, beta bursts, which are taken as evidence of the frontal (perhaps rIFG) signal to stop, only occur on ~15% of stop trials (14.6% in Experiment 4 and 16.2% in Experiment 5). Additionally, beta bursts often occur on go and stop-failure trials, sometimes at similar rates to successful stop trials (e.g., 15.4% in Experiment 5 for stop-failure rates). Therefore, beta bursts happen on few stop trials and almost as many go trials as stop trials. How does this signal that is neither necessary or sufficient for stopping explain stop success generally? Might there be differences between stop success trials with and without beta bursts? Could trials with beta bursts be edge cases of some sort (e.g., when proactive control is low and reactive stopping is therefore particularly essential) that are not indicative of the general mechanism for stopping on the other 85% of stop success trials?

Theoretical Implications:

13) I agree with the authors that these results have potentially striking implications for existing models of stopping. However, I think the paper could be improved by laying out these implications more explicitly. For instance, how does this framework relate to existing models including the original Independent Race Model (Logan and Cowan, 1984 / Psych Review), updated Independent Race Models including blocked input models (Logan et al., 2014; Logan et al., 2015), the Interactive Race Model (Boucher et al., 2007), the BEESTs model and trigger failures (Matzke et al., this relates to comment 5), and perhaps recent work suggesting violations of independence that can be accounted for by variable, sometimes weak inhibition (Bissett, Poldrack and Logan, in revision https://psyarxiv.com/kpa65).

[Editors' note: further revisions were suggested prior to acceptance, as described below.]

Thank you for submitting the revised version of your article "Temporal cascade of frontal, motor and muscle processes underlying human action-stopping" for consideration by *eLife*. Your revised article has been reviewed by two peer reviewers. The reviewers have discussed the reviews. The evaluation has been overseen by a Reviewing Editor (Wery van den Wildenberg) and Richard Ivry as the Senior Editor. The Reviewing Editor has drafted this decision to help you prepare a revised submission.

The thorough review reports of the previous review round match the vast nature and the wide scope of the original manuscript. The resulting rebuttal letter that accompanied the revision also reflects a broad discussion, which is commendable. Concerning the revised manuscript, there are a few issues left to be discussed.

These issues are addressed in detail at the end of this letter. Here I provide a brief summary:

The first remaining issue concerns the request to not merely acknowledge connecting studies by citing them, but to also address and integrate the findings of these relevant studies vis-a-vis the current work. In this respect, the rebuttal letter provides a more balanced account, and that tone might be extended to the manuscript.

The second issue concerns the timing when stopping reaches the muscle in relation to the model. This point has been brought to your attention by the Senior Editor, a few days before this decision letter, to facilitate the exchange process. Below you can find the whole line of reasoning (main point 2). More data, additional analyses or simulation studies are not required (or even asked) at this point. A discussion of the seemingly controversial time-courses would be sufficient.

Elaboration on these issues:

1) The authors wrote a detailed and insightful response, which essentially clarified my methodological questions. Some of the more conceptual issues were answered well in the response letter, yet the changes in the manuscript were less substantial than what one might have expected.

a) E.g., regarding Wessel's beta-burst analysis: the authors now cite this work (Results), but rather casually so. The manuscript does not try to compare or integrate the presented findings with Wessel's.

b) The same goes for the partial EMG work. Most of the studies we provided in the first round of the review are cited, but again rather offhand. The manuscript does not really integrate the different studies. Several studies already used an estimate essentially the same to what the authors still refer to as "our idea of CancelTime" (although admittedly these claims have been toned down), yet no attempt is made to compare these estimates, their correlations with SSRTs etc.

c) In some ways, I find that some of the claims made are not founded well enough in data, and the authors seem to be aware of it: "we agree that the temporal cascade model that we have constructed is only approximate as it is averaged across participants who had different stopping latencies, and is also contingent on, as the reviewers' rightly point out, on statistical procedures."

Yet again, the manuscript has not really been changed in accordance with this. Still, it is not properly discussed, for example, that the exact timing estimates heavily rely on the choice of latency estimates (EMG/MEP onsets or peak, of which each again can be calculated in many different ways.

2) I reviewed the initial submission of this manuscript. The manuscript is significantly improved in many ways. The editor requested that we evaluate whether claims have been toned down, previous work on EMG and beta-bursts have been acknowledged, and main point 6 regarding the timing estimate of CancelTime and a ballistic phase have been addressed. I think the first two points have been sufficiently addressed. However, the authors response to Main Point 6, especially their point 3, has raised new concerns about their claims about timing of stopping.

The authors present a 3-part response to main point 6. Part 1 points out that SSRTemg is similar to CancelTime and argues that this is consistent with the remaining ~60ms difference between CancelTime and SSRTbeh being a ballistic stage. Part 2 presents simulations consistent with a ballistic stage that is 34-47ms long (which is perhaps less than the ~60ms difference between canceltime and SSRTbeh). Point 3 argues that the real end of the race (at least as a criterion for evaluating CancelTime) is when EMG amplitude exceeds the threshold set on PartialEMG trials. They show that on correct go trials this threshold is exceeded ~100ms after the average stop signal would occur. Given the SSD tracking algorithm that ensured that the race between going and stopping is roughly tied, then the real latency of the stop process (as measured by EMG) may be ~100ms, so CancelTime may actually be an overestimate of the average stop latency.

The part 3 simulation results seem to conflict with multiple pieces of evidence in the manuscript and the literature. It appears that the authors are suggesting that the ballistic stage may be ~120ms (~220ms SSRTbeh minus the ~100ms stop process), which is both inconsistent with the simulation results in part 2 and inconsistent with the previous literature presented in the manuscript (e.g., Gopal and Muphy, 2016; Jana and Murthy, 2018 suggested 50ms for reaching movements). It also seems to bring into question the entire temporal sequence presented in the manuscript. Why would the end of the race be observable in muscles at 100ms if the signal to stop from cortex (perhaps rIFG) occurs 20ms later at 120ms?

Additionally, none of the 3 parts of the response have directly addressed the reviewers' main point 6. To briefly reiterate, assuming a race model, the ~50% of trials that are true failed stops (an overt response occurs) should tend to have the longest SSDs, fastest go RTs, and slowest SSRTs. The ~25% of trials that are successful stops with no EMG should be the opposite: shortest SSDs, slowest go RTs, and fastest SSRTs. This leaves the ~25% of trials that are successful stops with EMG, which should be in between these extremes. However, because there are ~twice as many stop failures as stop successes, the partial EMG trials will tend have shorter SSDs, slower go RT, and faster SSRTs than each measure's overall average. Therefore, CancelTime may be an underestimate of the true central tendency of SSRT across all stop trials.

In the text, the authors seem to agree with parts of this point, at least in part. They say "CancelTime… does not include the stopping latencies of the No EMG trials, which presumably reflect the fastest stopping latencies where the Stop process was fast enough to cancel the impending response before it reaches the muscle". However, they do not point out that CancelTime also does not include stop-failure trials, which presumably reflect the slowest stopping latencies, leaving CancelTime to reflect stopping latencies that are faster than the slowest half of stop trials but slower than the fastest quarter of stop trials. Also, in Figures 1G-I in their response, they illustrate how they believe correct stops with No EMG, correct stops with partial EMG, and failed stops arise from an accumulator model framework. No EMG has the fastest stop process, failed stop has the slowest stop process, and partial EMG is in between.

To conclude, I do not think that the response to main point 6 addressed the original concern, and I believe that the new simulations bring up new questions about the temporal cascade of processes in stopping. Does the stop process reach the muscle ~150-160ms after the stop signal, as suggested by CancelTime and SSRTemg, or is it ~100ms after the stop signal, as seemingly suggested by their part 3 in response to Main Point 6? If the latter, then how does this fit in with TMS evidence of motor suppression at ~140ms or beta-burst in cortex (perhaps rIFG) at 120ms? Also, if the argument from Main Point 6 is valid (and the authors do not address its validity directly), how can this be synthesized with the Part 3 response suggesting that CancelTime is an overestimate of the latency of the stop process?

---

## [Author Response]

Essential revisions:Relevant Background Literature:1) The authors are kind to mention work by Huster et al. on residual or partial EMG in successful stop trials. It should be noted though that earlier work already went into a similar direction yet does not seem to receive enough credit (in general, and not specifically with respect to this manuscript). The authors may thus consider including references to this earlier work. Some examples:- de Jong et al., 1990.- McGarry (1999). On the nature of stopping a voluntary action. Dissertation, University of British Columbia, Canada. (see also McGarry's articles from 2003 in Motor Control and Q J Exp Psychol).- Goonetilleke et al. (2010). A within-trial measure of the stop signal reaction time in a head-unrestrained oculomotor countermanding task. J Neurophysiol.- And last but not least, some more recent articles/manuscripts:- Atsma et al. (2018). Active Braking of Whole-Arm Reaching Movements Provides Single-Trial Neuromuscular Measures of Movement Cancellation. J Neurosci.- Raud et al. (submitted). Differences in unity: the go/no-go and stop signal tasks rely on different inhibitory mechanisms. bioRxiv 705079; doi: https://doi.org/10.1101/705079- Huster et al. (submitted). The P300 as marker of inhibitory control – fact or fiction? bioRxiv 694216; doi: https://doi.org/10.1101/694216- Thunberg et al. (submitted). Stimulating stopping? Investigating the effects of tDCS over the inferior frontal gyri and visual cortices. bioRxiv 723296; doi: https://doi.org/10.1101/723296- In light of this relatively rich prior history, the authors may want to consider to tone down their claim on this approach (E.g., "We predict that our new single-trial method of CancelTime will be more sensitive than SSRTBeh.").

The reviewers are correct to point out that there is a long history of studies evaluating EMG in the context of stopping. We now better acknowledge all of the above-mentioned papers in the revised manuscript and we have toned down our claims over ownership of the approach (Introduction, and Discussion).

SSRTEMG measure of Stopping:2) I am admittedly torn when it comes to the SSRTEMG. On the one hand, one may argue that it could avoid rather unspecific biases associated with the use of different response systems for the estimation of the SSRT. Then again, whereas plausible, it is not really clear that this is the case as little research systematically assessed the influence of response devices on the shape and timing of the EMG. Also, I see no good justification for classifying partial response EMG trials as failed stop trials; no overt response was produced, which is the instruction given to the participants, and I think there is no good justification to only classify trials without any EMG response as successful stop trials. In the end, I am unsure about how to interpret this measure meaningfully, and whether it can be used to "validate" the CancelTime (as is implied here). Not least, the calculation of the SSRTEMG relies on a rather small subset of SSDs (three per subject), as described in the subsection “Data analysis”, whereas usually a wide sampling of SSDs is recommended.

First, we would like to sincerely apologize for an erroneous reporting in the Materials and methods section of the manuscript, *all* SSDs were considered for calculation of SSRT_EMG_, and *not* 3 SSDs per subject. The results mentioned for SSRT_EMG_ thus considered the whole range of SSDs per subject. In fact, when the 3 most frequent SSDs are selected, mean SSRT_EMG_ = 163 ± 6 ms, as opposed to 157 ± 7 ms in the manuscript. We have accordingly changed this in the revised manuscript (subsection “Data analysis”).

The reviewers point out that the use of SSRT_EMG_ could potentially avoid biases in the determination of stopping latency associated with different response devices, but question whether this is likely to be the case because we don’t know how EMG profiles and onsets differ across devices. However, SSRT_EMG_ relies mainly on being able to detect an increase in EMG above baseline, and not really on the shape of the EMG after onset. EMG onsets can be reliably detected (Gopal et al., 2015; Jana et al., 2016) and correlate well with response times across a range of response devices (Botwinick and Thompson, 1966) (Figure 1—figure supplement 1), despite differences mechanical responses and thus electromechanical delays, supporting their use as a criterion measure. Thus, for simple movements (e.g. single joint) one might expect EMG onsets, but not electromechanical delays, to remain relatively consistent across devices, and consequently that SSRT_EMG_ might be less susceptible to device-related differences in estimations of stopping latency.

Moreover, our contention is that a large part of the difference between SSRT_Beh_ and the time of EMG cancellation (CancelTime) is due to an inherent “ballistic stage” in movements, and once the muscle activity crosses the point-of-no-return they can no longer be stopped, and a movement is inevitable (De Jong et al., 1990; Mirabella et al., 2006; Osman et al., 1986; Verbruggen and Logan, 2009). The duration of such ballistic stages has been estimated to be 10-25 ms in saccades in non-human primates (Boucher et al., 2007; Kornylo et al., 2003; Purcell et al., 2010) and 40-50 ms for reaching movements in humans (Gopal and Murthy, 2016; Jana and Murthy, 2018). In other words, the time of EMG cancellation on partial trials reflects a time just before the point-of-no-return, whereby if EMG activity is allowed to continue beyond this point it will exceed a critical threshold such that a button press necessarily ensues (we presume this threshold reflects the point at which the inertia of the finger is overcome). In this respect, what is being tracked by the SSD staircasing procedure is the probability of crossing that EMG threshold, but since SSRT is calculated based on keypress response times, it inevitably incorporates the ballistic stage that follows the crossing of this threshold. The purpose of the SSRT_EMG_ estimation was to test the idea by essentially removing the influence of any potentially ballistic phase (incorporating any electromechanical delays and inertia in the neuromuscular system and response device), on the estimated stopping latency. In order to do this, we re-classified successful and failed stops on the basis of the absence or presence of an EMG activation. The results of this analysis produce an estimate of stopping latency that very closely approximates our estimate based on CancelTime, and that are considerably shorter than the button-based estimate of SSRT. We have now added a few lines explaining the rationale in the revised manuscript (Results).

We admit that this was a post hoc analysis of data and that the task was designed to titrate SSD based on button presses, and not on the basis of whether EMG was detected. However, we believe it is reasonable to classify a response as the presence or not of EMG. Although participants were not explicitly trying to prevent EMG from being emitted, ~50% of successful stop trials contained no EMG, and so it seems reasonable to ask how quick the stopping latency has to be to prevent any EMG from being emitted. Also, in the wider context of action stopping, the choice of what is considered a response is rather arbitrary and depends on the question at hand. For example, whilst most studies, including ours, tend to use button presses, others have used continuous force or displacement measures and defined a response as anything exceeding a fixed force (25% maximum, (De Jong et al., 1990)), displacement threshold (0.01 m radius, (Atsma et al., 2018)) or velocity threshold (10% of peak velocity, (Jana and Murthy, 2018)). Therefore, we feel that this analysis is still informative, though we now acknowledge in the manuscript that our analysis of SSRT_EMG_ would potentially benefit from further validation in future, basing the SSD staircasing on whether EMG was present or not (Results).

In effect, the SSRT_EMG_ measure also might provide a solution to a long-standing puzzle in the literature where global motor suppression, i.e. the time when the Stop is implemented in the brain, is seen at ~150 ms of the Stop signal (Cai et al., 2012; Wessel et al., 2013) while SSRT_Beh_, i.e. the actual stopping of the response,is ~ 60 ms greater. The difference between SSRT_EMG_ and SSRT_Beh_ suggests that this discrepancy might be due to a ballistic stage.

3) When testing the partial response EMG against different SSDs, the H0 of no difference is justified by means of "weak go processes". I am wondering though what that implies: a very slow response (i.e., shallow slope), or one with a high threshold (thus unlikely to cause a button press), or a relatively delayed onset?

By “weak Go process” we mean one in which the individual intended to execute a response but failed to reach the threshold level of muscle activity required to fully depress the button. We have clarified this in the manuscript (Results).

Functional interpretation of CancelTime:4) The authors stress repeatedly that the CancelTime can be considered or is used as single-trial measure of the stop process. I suggest to tone this claim down, because:- it is not really used as a single-trial measure in the vast majority of analyses (with exceptions being the SD-based correlations of CancelTime and SSRTBEESTS, or the CancelTime-MEP alignment).- a proper metric should undergo testing in terms of its psychometric properties (reliability, validity, objectivity etc.). This is too often overlooked, not only but especially with psychophysiological measures.Thus, whereas I am also hopeful that the CancelTime may serve us as such a measure one day, I think we should not make this claim yet.

We agree with the reviewers’ assessment that in the majority of the analyses in the manuscript the single-trial capability of CancelTime has not been utilized. We chose to mention this to highlight the possibility of using CancelTime as a single-trial measure in future experiments. Indeed, we are currently exploring the possibility of using CancelTime as a single-trial measure (see Hannah et al., 2019 for single-trial correlations with right frontal beta bursts), and using the shape of the CancelTime distribution to better understand the network underlying human action-stopping. Likewise, we admit that the measure has not undergone thorough evaluation of its psychometric properties. Therefore, in line with the reviewers’ suggestion, we have toned down such statements (Abstract and Discussion). However, as alluded to above we have recently submitted a new manuscript (Hannah et al., 2019), in which we use single pulse TMS over the right inferior frontal cortex while concurrently measuring CancelTime from the hand. We show that real vs. sham TMS elongates CancelTime, and especially in those participants for whom the time of the pulse came closer to Cancel Time. This was a double-blind study, and the Sham coil was roughly equally as uncomfortable. The fact that real TMS elongated CancelTime reaffirms that it is a metric of stopping, and the single-trial correlations with frontal beta bursts supports its use on a single-trial basis.

5) In order to validate CancelTime, the authors show high correlations with other stop task measures including SSRTbeh and BEESTs model estimates. Along with other traditional measures, CancelTime was positively correlated with trigger failure rate. This seems to challenge the discriminant validity of CancelTime (and/or trigger failure rate). CancelTime was calculated on trials in which a stop signal has been triggered, reaches the muscle of the hand, but a behavioral response is not emitted. It is also intended to be a purer measure of response inhibition. Trigger failures are putative failures of attention and should only occur on failed stop trials. I suggest that the authors discuss this relationship between CancelTime and trigger failure rate.

Trigger failure trials ostensibly reflect those trials where the participant failed to detect the Stop signal or initiate the Stop process (Band et al., 2003). Unfortunately, there is no established physiological metric of this detection (although the N1 ERP during auditory SST might hold some promise), hence we relied on the BEESTS output which provides a single estimate of the probability of observing trigger failures (*p*(TF)) in a participant. The relationship we observed between CancelTime and trigger failure rate might be due to several factors. First, the reviewers’ point relies on the assumption that the detection of the stop signal and implementation of the stop process are independent processes, but they might not be. In other words, it is possible that a participant who is more likely to fail to detect the Stop Signal, is also likely to be slower to initiate the stop process which would result in a relationship between *p*(TF) and CancelTime. Second, it is not entirely obvious that so-called “trigger failures” truly reflect a failure to trigger the Stop process. Another interpretation could be that the measure reflects the “strength” of the Stop process once activated, in other words activation of the Stop network (Band et al., 2003) [but see (Matzke et al., 2017)]. One could imagine that a stronger Stop process would be quicker to stop a response leading to a relationship between *p*(TF) and CancelTime. We have commented on these issues in the Discussion section.

Concerning Temporal Dynamics:6) If I am understanding correctly, CancelTime is computed exclusively from partial EMG trials. Additionally, it is intended to be an alternative to SSRTbeh, which is a measure of the central tendency of SSRT across all trials. When CancelTime is compared to SSRTbeh, CancelTime is ~60ms shorter. The authors argue that CancelTime is shorter than SSRTbeh because CancelTime does not include the (~60ms) electromechanical delay and therefore is a purer measure of stopping. However, I would like to suggest another explanation for why CancelTime is shorter than SSRTbeh.Assuming (1) a race model architecture, (2) variability in the go RT and (3) variability in SSRT, then failed stops will tend to occur when SSD is long, go RT is fast, and SSRT is slow. Successful stops will tend to occur when SSD is short, go RT is slow, and SSRT is fast. Partial EMG trials, which I understand to be trials in which the go process doesn't win as decisively as a successful stop trial or fail as decisively as a failed stop trial, should have SSD, go RT, and SSRT values between these two extremes. However, ~1/2 of trials are failed stop, ~1/4 of trials as partialEMG, and ~1/4 of trials are successful stops. Therefore, partialEMG trials should be preferentially sampled from the half of trials with shorter SSD, slower go RT, and faster SSRT. If this is the case, then CancelTime being faster than SSRTbeh may be partially or completely explained by CancelTime being selectively sampled from faster SSRT trials whereas SSRTbeh is computed across all stop trials.To concretize my suggestion in an example, if we assume an SSRT distribution with variability such that the first quantile has a mean of 120 ms, the second 160 ms, the third 250, and the fourth 350 ms. When the first quantile is sampled, these should preferentially result in successful stops, when the second is sampled these should preferentially result in partialEMG trials, and when either of the latter two are sampled these should preferentially result in failed stop. However, if you computed CancelTime on these data it should be closer to 160ms (as a result of preferentially sampling the 2nd quantile) but the overall central tendency of the SSRT distribution would be ~220ms.This alternative implies that the average neural latency for stopping may not differ from SSRTbeh, or may differ less than the ~60 ms suggested here. One tool for evaluating this alternative explanation may be to simulate stop trials with variability in go, stop and SSD and evaluate whether computing SSRT only (or preferentially) on trials sampled from the 2nd quartile of SSRT can produce significant differences from the traditional method of computing SSRTbehav. If it may be useful, here is some openly available code that instantiates the interactive race framework and would allow varying go, stop, and SSD: https://github.com/bissettp/SharingContextDependence/blob/master/interactive_race.ipynb).

The reviewers raise an interesting point, however, there are several reasons why we think our interpretation that the extra 60 ms reflects an electromechanical delay, rather than us sampling preferentially from the faster portion of the stopping distribution.

1) As we hopefully clarified in an earlier response, our calculation of SSRT_EMG_ produces an estimate of stopping latency that is similar to CancelTime. Since the SSRT_EMG_ calculation utilized all the SSDs it is unlikely that our results are a manifestation of preferential sampling of the faster half of the true Stop distribution, and instead suggests that the difference between the two measures is due to a ballistic stage in responses that is insensitive to the Stop process. We now better describe our rationale behind calculating SSRT_EMG_ in the revised manuscript (Results).

2) We further tested the idea that there might be a ballistic stage in responses which inflates SSRT_Beh_ by simulating the independent race model. Here, the Go and Stop accumulator raced to a threshold, and the outcome of the trial was determined by which accumulator reached the threshold first. The activity of each accumulator could be described by the mean drift rate (μ) and the SD of the drift rate (σ) (Boucher et al., 2007; Ramakrishnan et al., 2012; Usher and McClelland, 2001) [Please note that although the interactive race model might be more appropriate, we choose the independent race model for simplicity. Future studies might investigate these issues further]. First, we tested whether the independent race model without a ballistic stage could fit the empirical results. It provided a poor fit suggesting that a modification to the model is needed. We then added a ballistic stage to the model and observed a better fit indicating that the difference between CancelTime and SSRT_Beh_ might be a reflection of this ballistic stage. Below we describe our steps.

To test the independent race model without a ballistic stage, for each participant, we estimated the Go parameters, (μ_GO-EMG_, σ_GO-EMG_) and (μ_GO-Beh_, σ_GO-Beh_) which best fit the RT_EMG_ (Go cue to EMG onset) and RT_Beh_ (Go cue to keypress) distributions in the Correct Go trials, respectively [further details outlined in Appendix 1]. Next, using the estimated Go parameters (μ_GO-Beh_, σ_GO-Beh_), we estimated the Stop parameters (μ_STOP-Beh_,σ_STOP-Beh_) that best fit the behavioral inhibition function (Figure 1A). Similarly, for EMG, we used its respective Go parameters (μ_GO-EMG_, σ_GO-EMG_) to estimate the Stop parameters (μ_STOP-EMG_,σ_STOP-EMG_) using the EMG inhibition function (Figure 1D). We reasoned that the Stop distribution underlying the inhibition of both the EMG and keypresses should be the same as both are part of the same response, and thus μ_STOP-EMG_ should be similar to μ_STOP-Beh_. Additionally, the observed difference between the EMG and behavioral inhibition functions might be attributed to the difference in the underlying Go processes (RT_EMG_ vs. RT_Beh_). However, this was not the case. The Stop distributions estimated from the EMG and keypresses were distinct, where μ_STOP-EMG_ (188 ± 9 ms)was significantly less than μ_STOP-Beh_ (207 ± 5 ms; *t*(41) = 2.7, *p* = 0.011, *d* = 0.4, *BF_10_* = 3.7). Second, the difference between the two inhibition functions could not be attributed to just the difference between the two Go processes. To check this, we used (μ_STOP-Beh_,σ_STOP-Beh_) as the Stop distribution and (μ_GO-EMG_, σ_GO-EMG_) as the Go distribution, and simulated the EMG inhibition function. We observed that the simulated EMG inhibition function did not match the empirical one (Figure 1B). To quantify this, we compared the squared error between the simulated and empirical EMG inhibition functions to the squared error between the simulated and empirical behavioral inhibition functions. The difference was significantly greater for the EMG inhibition functions compared to the behavioral ones (Figure 1C; behavioral inhibition function, squared error = 0.10 ± 0.01; EMG inhibition function, squared error = 0.21 ± 0.02, *t*(41) = 4.8, *p* < 0.001, *d* = 0.7, *BF_10_* > 100). Conversely, when we used (μ_STOP-EMG,_ σ_STOP-EMG_) as the Stop distribution and (μ_GO-Beh_, σ_GO-Beh_) as the Go distribution, and simulated the EMG inhibition function we again did not observe a good fit (Figure 1E, F; EMG inhibition function, squared error between experimental and simulated data = 0.12 ± 0.02; RT inhibition function, squared error = 0.23 ± 0.02, *t*(41) = 4.7, *p* < 0.001, *d* = 0.9, *BF_10_* > 100). This incompatibility suggests that some change needs to be made to the model such that a single Stop distribution is able to fit both the EMG and behavioral inhibition functions. Hence, we made two changes based a model previously used to describe stopping of reaching movements (Gopal and Murthy, 2016; Jana and Murthy, 2018). First, we added a peripheral delay to the Go process reflecting the observed delay between RT_EMG_ and RT_Beh_. Second, we partitioned this delay period into a non-ballistic and ballistic stage. And we then tested whether this model with a ballistic stage fit the empirical results better than the one without a ballistic stage.

In this physiologically relevant model, the Go process comprises an accumulation phase and a delay phase. Once the accumulator hits the threshold, EMG responses can be observed, and then following a peripheral delay, the EMG builds up enough (i.e. the point-of-no-return) to be able to cross the inertia of the limb and generate a movement (Figure 1G). If the Stop reaches the threshold before the Go reaches the threshold, then no EMG is elicited (no EMG trial; Figure 1G). Further, if the Stop reaches the threshold early during the delay period when the EMG has not built up to a critical level (point-of-no-return) it will be able to inhibit the response resulting in a partial EMG trial (Figure 1H). However, if the Stop reaches the threshold after the EMG has reached the point-of-no-return a movement is inevitable, resulting in a Failed Stop trial (Figure 1I). Thus, the point-of-no-return partitions the delay phase into a non-ballistic (where Stop can act) and a ballistic stage (where Stop cannot act). We tested whether this model provides a better fit compared to the model without a ballistic stage. Indeed, this was the case, as incorporation of a ballistic stage allowed μ_STOP-EMG_ to fit the behavioral inhibition function much better (Figure 1J). We quantified this by comparing the squared error between the simulated and empirical behavioral inhibition functions (Figure 1K; Model with ballistic stage: squared error = 0.14 ± 0.02; Model without ballistic stage: squared error = 0.23 ± 0.02; *t*(41) = 4.2, *p* < 0.001, *d* = 0.7, *BF_10_* > 100). Across participants, the mean ballistic stage was estimated to be 34 ± 4 ms. Thus, based on our model, there exists a ballistic stage in keypress responses, which we think is responsible for the difference between CancelTime and SSRT_Beh_ (we now mention this in the Results, and add Figure 3—figure supplement 1 to the revised manuscript, and have added the simulation methods and results as an Appendix 1). While this measure was less than our estimate of ~60 ms, we note that some participants did not have a sigmoidal EMG inhibition function leading to suboptimal parameter estimation. Indeed, when we considered only those participants who had lower than the population median squared error, the duration of the ballistic stage was 47 ± 3 ms which is closer to our estimate of ~60 ms.

3) In a similar vein, we carried out another analysis which provides a similar answer, and is based on the idea that there is a critical EMG threshold that once exceeded will inevitably result, after some electromechanical delay, in a button press (i.e. a ballistic stage). Consider Author response image 1 showing theoretical distributions of the end of the Go and Stop processes assuming that button presses reflect the end of the Go process (solid lines). Due to the SSD staircasing procedure employed in the studies, on average, the Go and Stop processes should end at the same time. By contrast, on the Failed Stop trials the Go process ends earlier than the Stop process. Going by definition of SSRT, on average, the Go process across all Go trials ended (i.e. button was pressed) ~220 ms after the stop signal (computed using the keypress RT minus the mean SSD). Now note that our mean CancelTime preceded SSRT by ~60 ms. At first glance, this might be taken to suggest that the reviewer is correct and CancelTime samples from the left half of the stop end time distribution. However, such comparisons are unfair because the criterion for the end time is different in each case: button press versus EMG.

If there is a ballistic stage, then the end time of the Go process ought to be the time at which EMG crosses the putative threshold. We therefore estimated the end time of the Go process on Failed Stop trials and Go trials as the time at which EMG activity exceeded the mean of the EMG amplitude recorded on Partial trials (as our estimate of the threshold for a response, see Author response image 1). We expressed this relative to the time of the Stop signal for Failed Stop and Go trials, respectively. When we do this, the mean end time of the Go process for the Correct Go trials is ~100 ms, while CancelTime is ~150 ms. While these timings are only rough estimates, it suggests that, as the reviewers propose, CancelTime only samples from a part of the whole Stop distribution, *but* it does not seem to be sampling from the faster half. If anything, this analysis suggests it samples the slower half of the Stop distribution. This makes sense as CancelTime does not include the stopping latency in the No EMG trials, where the Stop process was presumably quick (or the Go process was slow). This also lines with our TMS evidence where we detect MEP suppression (ostensibly marking the implementation of the Stop process) ~30 ms prior to CancelTime. Thus, we believe that while CancelTime is much shorter than SSRT_Beh_, it is still an overestimation of the true stopping latency. We note this in the revised manuscript (Discussion).

Thus, taken together, these results provide additional evidence that, 1) CancelTime preferentially subsamples the true Stop distribution, but in contrast to the reviewers’ suggestion, we think that it likely samples from the center or potentially the slower half of the true Stop distribution; and, 2) The difference between CancelTime (and SSRT_EMG_) and SSRT_Beh_ might be due to a ballistic stage.

**Author response image 1. respfig1:** End time of the Stop process using keypress and EMG amplitude. (**a**) Schematic of the distributions of the mean end time of the Go and Stop process in the Correct Go (green), Failed Stop (orange), and the Successful Stop (red) trials, while considering the keypress as the response. The CancelTime distribution is represented in brown. Note that CancelTime is an EMG measure and preceded SSRT_Beh_ calculated using keypresses. (**b**) Same as (**a**) but here the response is considered as the time when the EMG reaches the mean amplitude of that in the partial EMG trials. Note that the all distributions other than CancelTime shift to the left. (*Inset*) Mean EMG amplitude in the Failed Stop (orange) and partial EMG (brown) trials in Study 1. (**c**) Histogram showing the distribution of the Failed Stop and Correct Go responses when considering response as the time when the EMG in these trials cross the mean amplitude of that in the partial EMG amplitude. Histogram of CancelTime is shown on the inverted y-axis for better visualization. The triangle and cross-hairs represent the mean ± s.e.m. Colors are the same as in (**a**). (**d**) Same as (**c**) but for all participants in Study 1 and 2. The histogram of the population is shown at the top. Note that CancelTime probably overestimates the mean of the true Stop distribution.

7) The authors stress that parts of their model of the processing sequence is supported by the fact that the temporal distance between cortical reduction in excitability (TMS) and the CancelTime corresponds to the corticospinal conduction time (here about 23 ms). The difference between the CancelTime and the reduced motorcortical excitability is about 15 ms (155 – 140). However, the onset of decreased motorcortical excitability, which would best correspond to the onset of the decline in MEP amplitude, does start before that, as can be seen in Figure 4A. An only somewhat smaller effect is apparent already at 120ms, with an onset probably even before that. This would add another 20+ ms to the equation, thus contrasting 23ms vs. 35+ms. The point I want to make is that, even though I find the interpretation of the findings interesting and plausible, care has to be taken not to over-interpret these measures since they are heavily dependent on our statistical procedures (sample size, size of presumably relevant effect, parameterization of onset times etc.).

Firstly, we would like to apologize as there was a minor error in the reporting of the data, and the actual mean CancelTime was 160 ms, not 155 ms as reported previously. The difference was due to slightly different cut-off and outlier rejection procedures compared to the other experiments. These procedures are now consistent across experiments. We have rectified this in the revised manuscript (Results, and in Figure 4B).

To the reviewers’ point, we agree that the temporal cascade model that we have constructed is only approximate as it is averaged across participants who had different stopping latencies, and is also contingent on, as the reviewers’ rightly point out, on statistical procedures. For example, there is a degree of uncertainty even in our estimates of corticomotor conduction latency. The reason for this is that the conduction delay measured by TMS-evoked MEP is biased towards assessing the fastest corticospinal pathways: large-diameter, fast-conducting corticospinal neurons with mono-synaptic connections to the spinal motoneurons (Day et al., 1989; Edgley et al., 1997; Groppa et al., 2012). Yet conduction velocities in the (primate) pyramidal tract vary hugely (Firmin et al., 2014) and we presume that voluntary motor commands recruit a mixture of both faster and slower conducting (i.e. smaller diameter neurons and/or poly-synaptic connections to the spinal motorneurons) pathways. This would mean our estimate of conduction time is an underestimate and that the mean time at which changes in motor cortical output are observed at the level of the muscle (i.e. in the electromyogram) is probably several milliseconds longer than our estimate of 23 ms. In line with the reviewers’ comments, we have acknowledged a degree uncertainty in our temporal model on the whole, as well as in relation to this particular issue concerning the timing of the MEP and EMG suppression (Results, Discussion).

8) The validity of the reduced TMS-MEPs as indicators of the stopping process is derived from the strong reduction in successful stop relative to go trials around 140 ms, a notion further supported by Figure 4D. It is not discussed, however, that MEPs also are reduced for unsuccessful stop trials around 140 ms locked to the stop signal (a reduction similar to that of successful stop trials; Figure 4B; the effect does not seem to be that much smaller). How to reconcile these findings?

We thank the reviewers’ for making this observation. Indeed, we do see a relative reduction in MEPs in the Failed Stop trials. The fact this happens doesn’t invalidate our use of the MEPs as an indicator of the Stop process. In fact, it seems to make some sense if we think that the Stop process is initiated (nearly) every time a Stop signal is presented (as the signal is quite salient and the studies were conducted on young, healthy adults), and that failure to stop generally occurs when the Go process is particularly fast or the Stop process is particularly slow. Thus, the MEP reduction in Failed Stop trials is the expression of the Stop process, i.e. it crossing the line, despite the fact that the Go process has already crossed the point-of-no-return. Indeed, the reviewers pointed out that in Successful Stops the MEP suppression might even begin a little earlier than in Failed Stops (120 ms versus 140 ms), which would be consistent with the idea that some stops fail because of a slower Stop process (rather than a particularly fast Go). Bear in mind too, that the suppression we are observing here is in a task-irrelevant muscle. A benefit of this is that we can observe the timing of the Stop-related suppression in the absence of any Go-related muscle activity that would otherwise overlap and potentially blur the true timing. We have included a few lines in the manuscript discussing this issue (Results).

9) In the fourth paragraph of the Discussion, the authors cite papers showing that in monkey, rats, and humans that neurons that are putatively related to stopping change close to SSRT. This seems like considerable evidence against the chronometric framework presented here that assumes that the stop command is sent from brain to muscle well before SSRT (~60 ms + conduction time). I think it would be helpful to understand the authors' position on how these seemingly contradictory findings relate to this current framework.

Although we acknowledge that additional evidence may be required, based on our current results, we believe that some of the brain signatures previously reported might not be causally-related to the initiation/implementation of the Stop per se [for example, the intraparietal sulcus which has been described to be important for stopping movements (Osada et al., 2019) might not be so (Hannah and Jana, 2019)]. They could instead reflect some other aspect of the Stop, for example, monitoring or feedback related to the implementation of the Stop as has been ascribed to certain brain signatures that modulate after SSRT (Logan et al., 2015; Schall and Boucher, 2007). We now discuss this in the revised manuscript (Discussion).

Relatedly, a recent study in non-human primates performing the Stop signal task has demonstrated that the premotor cortical network dynamics in the Successful Stop trials diverge from that seen in the Failed Stop and Go trials within 70-120 ms of the Stop signal (Pani et al., 2019). This again highlights that rapid action-stopping might be implemented in the motor system much earlier than that observed at the behavioral end point.

Functional Significance of beta Burst Activity:10) How does the work presented in Study 4 relate to the recent preprint of a paper by Wessel on the role of beta bursts in stopping? [Wessel, J.R., "β-bursts reveal the trial-to-trial dynamics of movement initiation and cancellation", https://www.biorxiv.org/content/10.1101/644682v1].

There are some clear similarities between the Wessel, 2019 study and ours. First, similar to what we have reported, he observes an increase in bursts% in frontocentral electrodes prior to SSRT in the Successful Stop trials compared to the Failed Stop trials. Second, this increase seems to occur quite early after the Stop signal and much before the SSRT (Figure 2B, 25-75 ms and 75-125 ms bin). In fact, the difference between the Successful and Failed Stop trials seems to subside close to SSRT (see Figure 5—figure supplement 4 for the burst% modulation across time). Thus, the average timing of beta bursts in the Successful Stop trials might be similar in both studies. However, we would also like to highlight some differences between the two studies. First, while Wessel analyzed the EEG channel space, we analyzed beta bursts in a frontal spatial filter. Indeed, the signal-to-noise observed in a spatial filter is much improved (Muralidharan et al., 2019; Wessel, 2018). This allowed us to detect beta bursts despite having far fewer number of subjects (N = 11, 13 vs. N = 234). In fact, we were unable to detect enough beta bursts in the channel space in our pool of subjects at all to make any predictions about the timings of the bursts (see Author response image 2 for exemplar subject where beta burst probability is computed over the entire channel space as in Wessel’s case). Second, while we defined bursts based on the beta power during a baseline period prior to the presentation of the Stop Signal, Wessel defined it based on the beta power in a channel across time. Third, Wessel does not show if the increase beta burst% in the Successful Stop trials prior to SSRT is greater than that observed before the Stop signal. Indeed, if beta bursts were to play a role in movement cancellation then one might expect it to increase once the Stop signal is presented. Thus, taking all these together, we believe that our method might be better suited for detection of beta bursts in future EEG studies. Nevertheless, despite all these differences, the study by Wessel, 2019, reinforces our observation that the timing of beta bursts in the frontal cortex might have a key role during movement cancellation.

**Author response image 2. respfig2:** Beta burst analysis in the channel space. Please note that the SNR is far worse than that observed using a spatial filter.

11) Study 4 finds an increase in beta bursts for successful stops relative to baseline and go trials, an effect interpreted as indicator of the engagement of the right inferior frontal stopping processes. This is a very interesting finding, but its interpretation is not without some conceptual problems.- Given that EEG is plagued with the inverse problem, it seems somewhat unjustified to claim that this signature originates in the right frontal cortex. But even if one would agree that a rough deduction of the source from its topography is possible, I would have a hard time to associate the rather midfrontal topography with a right frontal source close to the sylvian fissure. Beyond, the authors performed dipole fitting on the independent component topographies, and thus could easily check if the individual dipoles of these components at least roughly are fitted to such a position.

While we have converging evidence that beta bursts might have originated from the right frontal cortex (Swann et al., 2009, 2012), we agree that it is inappropriate to establish the origin of the beta bursts from EEG due to its poor spatial resolution. However, as suggested, we performed a dipole localization analysis of the right-frontal ICs which were selected from both Study 4 and Study 5 and plotted the average dipole location along and its variability in an average MNI space. The dipole is fronto-central (but biased towards the right) which roughly estimates the origin of this beta from the mid to right frontal cortex. However, future studies are required to establish whether the beta bursts recorded at the scalp do originate from the right inferior frontal gyrus, a key node in the stopping network, or rather to preSMA or to both (note that these are connected via white matter (Catani et al., 2013; Swann et al., 2012)). We discuss this in the revised manuscript (Discussion, and have added Figure 5—figure supplement 2 to the revised manuscript).

Beta burst activity is depicted and analyzed relative to the pre-stop period. If we leave the IC-selection aside for a moment, the resulting time courses may still be in accordance with an effect akin to motor beta: around the time of the stop signal, beta activity would be similarly low for all three categories due to motor preparation, followed by rebound-effects that might be earlier in case of successful stops (cancelled), succeeded by unsuccessful stops and go trials (rebound after execution).The IC selection states, however, that a component needed to show a beta power increase between the stop signal and the SSRT compared to a time window prior to the go cue. This should exclude potential motor beta components with typical time courses delineated above. The time course for the go activity supports this assumption. Nevertheless, given the conceptual importance of this beta component in this context, it would be nice to also see the go-locked time courses for the stop-related conditions.

We thank the reviewer for raising this issue. While it is possible that the motor beta rebound might contaminate the increase in burst% following the Stop signal, we think that this is unlikely to explain our results. First, as mentioned above, the dipole localization does not support a sensorimotor source for the generation of this beta and points to a frontal source. Secondly, post-movement beta rebound occurs at least ~500 ms after the EMG offset. However, in the frontal spatial filter, when the burst% is time-locked to the Go cue (see Figure 5—figure supplement 4), the increase in burst% in the Successful Stop trials starts well before both CancelTime (average time when EMG activity starts decreasing) and SSRT_Beh_ (average time of movement execution). In addition, there is either no modulation of burst% in the Go trials (Study 5) or the modulation occurred after the average RT (Study 4). We mention this in the revised manuscript (Results) and add Figure 5—figure supplement 4 to the revised manuscript).

12) I understand the presented beta burst data to be the probability that a beta burst occurred on any given trial. Assuming this, beta bursts, which are taken as evidence of the frontal (perhaps rIFG) signal to stop, only occur on ~15% of stop trials (14.6% in Experiment 4 and 16.2% in Experiment 5). Additionally, beta bursts often occur on go and stop-failure trials, sometimes at similar rates to successful stop trials (e.g., 15.4% in Experiment 5 for stop-failure rates). Therefore, beta bursts happen on few stop trials and almost as many go trials as stop trials. How does this signal that is neither necessary or sufficient for stopping explain stop success generally? Might there be differences between stop success trials with and without beta bursts? Could trials with beta bursts be edge cases of some sort (e.g., when proactive control is low and reactive stopping is therefore particularly essential) that are not indicative of the general mechanism for stopping on the other 85% of stop success trials?

Indeed, it is quite curious why beta bursts are observed on a small percentage of trials. One potential explanation could be that we are not able to detect a burst reliably on every trial. This could be due to the low SNR of EEG and if we were recording directly from these brain regions, we could get a better estimate of the percentage of bursts in Successful Stop trials. However, the presence of beta bursts in the Go trials does raise a question about the role of beta bursts. It is possible that beta bursts are (partly) spontaneous events that occur all the time (but have some functional consequence) (Shin et al., 2017), or it might have a role in proactive slowing in the Go trials (as they are embedded in a task where the participants have to stop their response in a minority of trials). Nevertheless, we did an additional analysis of looking at CancelTimes in the Successful Stop trials with and without bursts between Stop signal and SSRT_Beh_. There was no difference in the CancelTimes between these trial types (CancelTime_With Burst_ = 164 ± 9 ms; CancelTime_No Burst_ = 165 ± 9 ms, *t*(12) = 0.57, *p* = 0.58, *d* = 0.2, *BF_10_* = 0.32) We now discuss this in the revised manuscript (Discussion).

Theoretical Implications:13) I agree with the authors that these results have potentially striking implications for existing models of stopping. However, I think the paper could be improved by laying out these implications more explicitly. For instance, how does this framework relate to existing models including the original Independent Race Model (Logan and Cowan, 1984 / Psych Review), updated Independent Race Models including blocked input models (Logan et al., 2014; Logan et al., 2015), the Interactive Race Model (Boucher et al., 2007), the BEESTs model and trigger failures (Matzke et al., this relates to comment 5), and perhaps recent work suggesting violations of independence that can be accounted for by variable, sometimes weak inhibition (Bissett, Poldrack, and Logan, in revision https://psyarxiv.com/kpa65).

Our study sheds light on the temporal profile of the physiological/network model underlying action-stopping (Aron et al., 2014). To this end we characterize the chronometrics of the Stop process and observe, frontal beta activity at ~120 ms, followed by decreased M1 excitability at ~140 ms, and cancellation of the muscle response at ~160 ms. As we have not formally tested all the computational models existing in the literature, it is probably premature to comment on the computational relationship between the Go and the Stop process. However, we can conjecture as to which models might best explain our results. Our results are not compatible with a strictly independent model since we see active inhibition of M1 (the Go process) already some time before SSRT. We think that the interactive race model (Boucher et al., 2007) best explains out data, but we must note that this model was originally tested on inhibition of saccades on over-trained non-human primates (SSRT ~90 ms), whereas our results are on inhibition of keypresses in naïve human participants (SSRT ~220 ms). Boucher et al. considered SSRT as a sum of Delay_Stop_ (delay in activation of the Stop process, ~60 ms, i.e. ~70% of the SSRT), Stop_interrupt_ (duration in which the Stop process is implemented, starting from the accumulation to the inhibition of the Go process, ~20 ms), and Go_Ballistic_ (ballistic stage preceding movement initiation, ~10 ms). First, frontal beta bursts occur ~120 ms after the Stop while CancelTime is ~160 ms, i.e. the Delay_Stop_ is ~75% of the duration of the Stop duration, which is similar to that reported by Boucher et al., 2007. Second, frontal beta bursts are followed by decrease in M1 excitability within ~20 ms. This is similar to the Stop_interrupt_ value reported by Boucher et al., 2007. However, the duration of the ballistic stage is probably longer for manual responses (~50 ms for reaching movements (Gopal and Murthy, 2016; Jana and Murthy, 2018)). Our simulations also support this (see response to a previous question). Thus, we believe that, SSRT_Beh_ = Delay_Stop_ + Stop_interrupt_ + Delay_Corticospinal_ + Go_Ballistic_.

The initial delay in activation of the Stop process might relate to the triggering/detection of the Stop process [which might take about 80-120 ms (Bekker et al., 2005)] which is a key component of the BEESTS model. Another support of the interactive race model is that, we observe a decrease in corticomotor excitability in task-unrelated muscles at ~140 ms which persists for at least ~60 ms. This probably reflects an active inhibition of the Go process to directly suppress the output neurons of the primary motor cortex, but we presume that there is also a suppression of upstream drive to the motor cortex. For example, exogenous suppression of motor cortical output with TMS over the primary motor cortex seems only to delay voluntary motor output, rather than abolish it (Day et al., 1989). Thus, while we favor the interactive race model where the initial delay is related to the triggering/detection of the Stop signal. We note that the interactive-race model and blocked-input model are very similar (Logan et al., 2015), so our results do not disambiguate them. We now discuss this in the revised manuscript (Discussion).

References:

Band GPHH, van der Molen MW, Logan GD. 2003. Horse-race model simulations of the stop-signal procedure. Acta Psychol (Amst) 112:105–142. doi:10.1016/S0001-6918(02)00079-3

Botwinick J, Thompson LW. 1966. Premotor and motor components of reaction time. J Exp Psychol 71:9–15. doi:10.1037/h0022634

Firmin L, Field P, Maier MA, Kraskov A, Kirkwood PA, Nakajima K, Lemon RN, Glickstein M. 2014. Axon diameters and conduction velocities in the macaque pyramidal tract. J Neurophysiol 112:1229–1240. doi:10.1152/jn.00720.2013

Gopal A, Viswanathan P, Murthy A. 2015. A common stochastic accumulator with effector-dependent noise can explain eye-hand coordination. J Neurophysiol 2033–2048. doi:10.1152/jn.00802.2014

Hannah R, Jana S. 2019. Disentangling the role of posterior parietal cortex in response inhibition. J Neurosci 39:6814–6816. doi:10.1523/JNEUROSCI.0785-19.2019

Jana S, Gopal A, Murthy A. 2016. Evidence of common and separate eye and hand accumulators underlying flexible eye-hand coordination. J Neurophysiol 117:348–364. doi:10.1152/jn.00688.2016

Matzke D, Hughes M, Badcock JC, Michie P, Heathcote A. 2017. Failures of cognitive control or attention? The case of stop-signal deficits in schizophrenia. Attention, Perception, Psychophys 79:1078–1086. doi:10.3758/s13414-017-1287-8

Muralidharan V, Yu XY, Cohen MX, Aron AR. 2019. Preparing to Stop Action Increases Beta Band Power in Contralateral Sensorimotor Cortex Vignesh. J Cogn Neurosci 31:657–668. doi:10.1162/jocn

Pani P, Giamundo M, Giarrocco F, Mione V, Brunamonti E, Mattia M, Ferraina S. 2019. Neuronal population dynamics during motor plan cancellation in non-human primates. bioRxiv 774307. doi:10.1101/774307

Wessel JR. 2018. Testing Multiple Psychological Processes for Common Neural Mechanisms Using EEG and Independent Component Analysis. Brain Topogr 31:90–100. doi:10.1007/s10548-016-0483-5

[Editors' note: further revisions were suggested prior to acceptance, as described below.]The thorough review reports of the previous review round match the vast nature and the wide scope of the original manuscript. The resulting rebuttal letter that accompanied the revision also reflects a broad discussion, which is commendable. Concerning the revised manuscript, there are a few issues left to be discussed.These issues are addressed in detail at the end of this letter. Here I provide a brief summary:The first remaining issue concerns the request to not merely acknowledge connecting studies by citing them, but to also address and integrate the findings of these relevant studies vis-a-vis the current work. In this respect, the rebuttal letter provides a more balanced account, and that tone might be extended to the manuscript.The second issue concerns the timing when stopping reaches the muscle in relation to the model. This point has been brought to your attention by the Senior Editor, a few days before this decision letter, to facilitate the exchange process. Below you can find the whole line of reasoning (main point 2). More data, additional analyses or simulation studies are not required (or even asked) at this point. A discussion of the seemingly controversial time-courses would be sufficient.Elaboration on these issues:1) The authors wrote a detailed and insightful response, which essentially clarified my methodological questions. Some of the more conceptual issues were answered well in the response letter, yet the changes in the manuscript were less substantial than what one might have expected.

We apologize for not integrating a more substantial discussion of these issues and hope that the new changes are much better.

a) E.g., regarding Wessel's beta-burst analysis: the authors now cite this work (Results), but rather casually so. The manuscript does not try to compare or integrate the presented findings with Wessel's.

We now include a more fulsome discussion of Wessel’s beta-burst study in the revised manuscript (Discussion).

b) The same goes for the partial EMG work. Most of the studies we provided in the first round of the review are cited, but again rather offhand. The manuscript does not really integrate the different studies. Several studies already used an estimate essentially the same to what the authors still refer to as "our idea of CancelTime" (although admittedly these claims have been toned down), yet no attempt is made to compare these estimates, their correlations with SSRTs etc.

We have removed the phrase “our idea”. We also discuss the previous studies in more detail (Discussion).

c) In some ways, I find that some of the claims made are not founded well enough in data, and the authors seem to be aware of it: "we agree that the temporal cascade model that we have constructed is only approximate as it is averaged across participants who had different stopping latencies, and is also contingent on, as the reviewers' rightly point out, on statistical procedures."Yet again, the manuscript has not really been changed in accordance with this. Still, it is not properly discussed, for example, that the exact timing estimates heavily rely on the choice of latency estimates (EMG/MEP onsets or peak, of which each again can be calculated in many different ways.

We agree with the reviewer that there is some uncertainty in our timing estimates. We did previously mention in the revised manuscript that the timings for the temporal model were approximate (Discussion), and as such Figure 6 is described as a “hypothetical” model of the temporal cascade of process underlying human action-stopping. We also previously had a section noting the imprecision in our estimates of MEP latencies (Results). We have now amended the manuscript to further acknowledge some of the likely sources of imprecision in our timing estimates (Discussion). We would like to point out too, that despite these uncertainties, our estimates do seem broadly consistent with those from numerous other studies, e.g. “CancelTime” (Hannah et al., 2019; Raud et al., 2019; Raud and Huster, 2017), time of MEP suppression (Coxon et al., 2006; van den Wildenberg et al., 2010), corticomotor conduction time (Groppa et al., 2012; Hamada et al., 2013), BurstTimes (Hannah et al., 2019), and the ballistic stage (Gopal and Murthy, 2016; Jana and Murthy, 2018). We now mention this explicitly in the revised manuscript (Discussion).

2) I reviewed the initial submission of this manuscript. The manuscript is significantly improved in many ways. The editor requested that we evaluate whether claims have been toned down, previous work on EMG and beta-bursts have been acknowledged, and main point 6 regarding the timing estimate of CancelTime and a ballistic phase have been addressed. I think the first two points have been sufficiently addressed. However, the authors response to Main Point 6, especially their point 3, has raised new concerns about their claims about timing of stopping.The authors present a 3-part response to main point 6. Part 1 points out that SSRTemg is similar to CancelTime and argues that this is consistent with the remaining ~60ms difference between CancelTime and SSRTbeh being a ballistic stage. Part 2 presents simulations consistent with a ballistic stage that is 34-47ms long (which is perhaps less than the ~60ms difference between canceltime and SSRTbeh). Point 3 argues that the real end of the race (at least as a criterion for evaluating CancelTime) is when EMG amplitude exceeds the threshold set on PartialEMG trials. They show that on correct go trials this threshold is exceeded ~100ms after the average stop signal would occur. Given the SSD tracking algorithm that ensured that the race between going and stopping is roughly tied, then the real latency of the stop process (as measured by EMG) may be ~100ms, so CancelTime may actually be an overestimate of the average stop latency.The part 3 simulation results seem to conflict with multiple pieces of evidence in the manuscript and the literature. It appears that the authors are suggesting that the ballistic stage may be ~120ms (~220ms SSRTbeh minus the ~100ms stop process), which is both inconsistent with the simulation results in part 2 and inconsistent with the previous literature presented in the manuscript (e.g., Gopal and Muphy, 2016; Jana and Murthy, 2018 suggested 50ms for reaching movements). It also seems to bring into question the entire temporal sequence presented in the manuscript. Why would the end of the race be observable in muscles at 100ms if the signal to stop from cortex (perhaps rIFG) occurs 20ms later at 120ms?Additionally, none of the 3 parts of the response have directly addressed the reviewers' main point 6. To briefly reiterate, assuming a race model, the ~50% of trials that are true failed stops (an overt response occurs) should tend to have the longest SSDs, fastest go RTs, and slowest SSRTs. The ~25% of trials that are successful stops with no EMG should be the opposite: shortest SSDs, slowest go RTs, and fastest SSRTs. This leaves the ~25% of trials that are successful stops with EMG, which should be in between these extremes. However, because there are ~twice as many stop failures as stop successes, the partial EMG trials will tend have shorter SSDs, slower go RT, and faster SSRTs than each measure's overall average. Therefore, CancelTime may be an underestimate of the true central tendency of SSRT across all stop trials.In the text, the authors seem to agree with parts of this point, at least in part. They say "CancelTime… does not include the stopping latencies of the No EMG trials, which presumably reflect the fastest stopping latencies where the Stop process was fast enough to cancel the impending response before it reaches the muscle". However, they do not point out that CancelTime also does not include stop-failure trials, which presumably reflect the slowest stopping latencies, leaving CancelTime to reflect stopping latencies that are faster than the slowest half of stop trials but slower than the fastest quarter of stop trials. Also, in Figures 1G-I in their response, they illustrate how they believe correct stops with No EMG, correct stops with partial EMG, and failed stops arise from an accumulator model framework. No EMG has the fastest stop process, failed stop has the slowest stop process, and partial EMG is in between.To conclude, I do not think that the response to main point 6 addressed the original concern, and I believe that the new simulations bring up new questions about the temporal cascade of processes in stopping. Does the stop process reach the muscle ~150-160ms after the stop signal, as suggested by CancelTime and SSRTemg, or is it ~100ms after the stop signal, as seemingly suggested by their part 3 in response to Main Point 6? If the latter, then how does this fit in with TMS evidence of motor suppression at ~140ms or beta-burst in cortex (perhaps rIFG) at 120ms? Also, if the argument from Main Point 6 is valid (and the authors do not address its validity directly), how can this be synthesized with the Part 3 response suggesting that CancelTime is an overestimate of the latency of the stop process?

We appreciate the insightful comments here and apologize for the confusion caused by our response. A key tenet of the paper, and our response to Main Point 6, is that SSRT may overestimate stopping latency by virtue of an inherent ballistic stage. All three of the additional analyses in Point 6 of our response provide further evidence of this: #1 the EMG-based calculation of SSRT very closely approximates CancelTime; #2 our simulation supported the notion of a ballistic stage, implying that SSRT_Beh_ might be an overestimate of the true stopping latency; and #3 used a crude estimate of the end of the Go process relative to a theoretical stop cue, and suggested the mean was ~100 ms, which is *closer* to CancelTime than it is SSRT_Beh_.

To be clear about #3, the idea we were attempting to illustrate with this analysis was that comparing the SSRT_Beh_ distribution with the CancelTime distribution is inappropriate because the former includes an electromechanical delay that the latter does not. The fact that the center of the Go distribution does not line up perfectly with the center of the CancelTime distribution probably reflects the particular EMG threshold chosen for the end of the Go process. This is a conservative estimate because the EMG amplitude on ~50% of the Partial EMG trials exceeded this value without generating an overt keypress, implying that there is actually a significant buffer. We note that raising the threshold slightly to account for this would bring the estimates of the end of the Go process closer to that of CancelTime.

We also note that unlike timing, the amplitude of EMG is quite variable and probably not the only determinant of response time (e.g. the rate of EMG rise as well as synergist and stabilizer muscle activity may contribute). This is evident in the fact that the tails of the EMG amplitude distributions for Partial EMG and Go trials overlap. Given these issues, we would urge against using this #3 analysis to infer the latency of stopping and thus do not feel that this analysis invalidates our temporal model of the processes contributing to stopping. We hope this allays the reviewer’s concerns.

In relation to the reviewer’s original point about whether CancelTime sub-samples from the true Stop distribution, our #1 response was intended to deal specifically with this issue. The point we were trying to make was that SSRT_EMG_ is calculated in the same manner as SSRT_Beh_, only the definition of a response differs. Thus, SSRT_EMG_ is calculated from the *whole* distribution, i.e. it does not sub-sample, and yet it still gives a similar estimate of stopping latency as CancelTime. This seems contrary to the reviewer’s suggestion that CancelTime underestimates the central tendency of the true Stop distribution.

In summary, our analyses provide converging evidence that SSRT_Beh_ seems to include a ballistic stage and thus may overestimate stopping latency. This creates a temporal disparity between SSRT_Beh_ and neurophysiological measures of stopping latency which appear much earlier. CancelTime measures the latency of stopping at the level of the muscle and bridges this temporal disparity. Thus, a main contribution of the paper is in demonstrating the temporal relationship between brain signatures and the muscle (CancelTime) and behavioral markers of stopping latency (SSRT_Beh_). [Another main contribution is that CancelTime is a potentially single trial estimate of the speed of stopping – which has big implications for our field].